# Smooth Kernels Improve Adversarial Robustness and Perceptually-Aligned Gradients

## Abstract

Recent research has shown that CNNs are often overly sensitive to high-frequency textural patterns. Inspired by the intuition that humans tend to be more sensitive to lower-frequency (larger-scale) patterns, we design a regularization scheme that penalizes large differences between adjacent components within each convolutional kernel. We apply our regularization onto several popular training methods, demonstrating that the models with the proposed smooth kernels enjoy improved adversarial robustness. Further, building on recent work establishing connections between adversarial robustness and interpretability, we show that our method appears to give more perceptually-aligned gradients.

## 1 Introduction

In recent years, deep learning models have demonstrated remarkable capabilities for predictive modeling in computer vision, leading some to liken their abilities on perception tasks to those of humans (*e.g.*, Weyand et al., 2016). However, under closer inspection, the limits of such claims to the narrow scope of i.i.d. data become clear. For example, when faced with adversarial examples (Szegedy et al., 2013; Goodfellow et al., 2015) or even in non-adversarial domain-agnostic cross-domain evaluations (Wang et al., 2019a;b; Carlucci et al., 2019), performance collapses, dispelling claims of human-like perceptive capabilities and calling into doubt more ambitious applications of this technology in the wild.

A long line of recent research has investigated the robustness of neural networks, including investigations of the high-dimension nature of models (Fawzi et al., 2018), enlarging the gaps between decision boundaries (Zhang et al., 2019a), training the models with augmented examples through attack methods (Madry et al., 2018), and even guaranteeing the robustness of models within given radii of perturbation (Wong & Kolter, 2018; Cohen et al., 2019). Compared to earlier methods, these recent works enjoy stronger robustness both as assessed via theoretical guarantees and empirically via quantitative performance against strong attacks. However, despite the success of these techniques, vulnerabilities to new varieties of attacks are frequently discovered (Zhang et al., 2019b).

In this paper, we aim to lessen the dependency of neural networks on high-frequency patterns in images, regularizing CNNs to focus on the low-frequency components. Therefore, the main argument of this paper is that: *by regularizing the CNN to be most sensitive to the low-frequency components of an image, we can improve the robustness of models. Interestingly, this also appears to lead to more perceptually-aligned gradients.* Further, as Wang et al. (2019c) explicitly defined the low (or high)-frequency components as images reconstructed from the low (or high)-end of the image frequency domain (as is frequently discussed in neuroscience literature addressing human recognition of shape (Bar, 2004) or face (Awasthi et al., 2011)), we continue with this definition and demonstrate that a smooth kernel can filter out the high-frequency components and improve the models' robustness.

We test our ideas and show the empirical improvement over popular adversarial robust methods with standard evaluations and further use model interpretation methods to understand how the models make decisions and demonstrate that the regularization helps the model to generate more perceptually-aligned gradients.

## 2 RELATED WORK

Adversarial examples are samples with small perturbations applied that are imperceptible to humans but can nevertheless induce misclassification in machine learning models (Szegedy et al., 2013)). The discovery of adversarial examples spurred a torrent of research, much of it consisting of an arm race between those inventing new attack methods and others offering defenses to make classifiers robust to these sorts of attacks. We refer to survey papers such as (Akhtar & Mian, 2018; Chakraborty et al., 2018) and only list a few most relevant works about applying regularizations to the networks to improve the adversarial robustness, such as regularizations constraining the Lipschitz constant of the network (Cisse et al., 2017) (Lipschitz smoothness), regularizing the scale of gradients (Ross & Doshi-Velez, 2018; Jakubovitz & Giryes, 2018) (smooth gradients), regularizing the curvature of the loss surface (Moosavi-Dezfooli et al., 2019) (smooth loss curvature), and promoting the smoothness of the model distribution (Miyato et al., 2015). These regularizations also use the concept of "smoothness," but different from ours (small differences among the adjacent weights).

Recently, adversarial training (Goodfellow et al., 2015; Madry et al., 2018) has become one of the most popular defense methods, based on the simple idea of augmenting the training data with samples generated through attack methods (*i.e.*, threat models). While adversarial training excels across many evaluations, recent evidence exposes its new limitations (Zhang et al., 2019b), suggesting that adversarial robustness remains a challenge.

*Key differences:* In this paper, we present a new technique penalizing differences among the adjacent components of convolutional kernels. Moreover, we expand upon the recent literature demonstrating connections between adversarial robustness and perceptually-aligned gradients.

## 3 SMOOTH KERNEL REGULARIZATION

**Intuition.** *High-frequency* components of images are those reconstructed from the high-end of the image frequency-domain through inverse Fourier transform. This definition was also verified previously by neuroscientists who demonstrated that humans tend to rely on the low-frequency component of images to recognize shapes (Bar, 2004) and faces (Awasthi et al., 2011). Therefore, we argue that the smooth kernel regularization is effective because it helps to produce models less sensitive to high-frequency patterns in images. We define a smooth kernel as a convolutional kernel whose weight at each position does not differ much from those of its neighbors, *i.e.*, $(\mathbf{w}_{i,j} - \mathbf{w}_{h,k \in N(i,j)})^2$ is a small number, where $\mathbf{w}$ denotes the convolutional kernel weight, $i, j$ denote the indices of the convolutional kernel $\mathbf{w}$, and $N(i,j)$ denotes the set of the spatial neighbors of $i, j$.

We note two points that support our intuition.

1. *The frequency domain of a smooth kernel has only negligible high-frequency components.*
   This argument can be shown with Theorem 1 in (Platonov, 2005). Roughly, the idea is to view the weight matrix $\mathbf{w}$ as a function that maps the *index of weights* to the weights: $w(i,j) \rightarrow \mathbf{w}_{i,j}$, then a smooth kernel can be seen as a Lipschitz function with constant $\alpha$. As pointed out by Platonov (2005), Titchmarsh (1948) showed that when $0 < \alpha < 1$, in the frequency domain, the sum of all the high frequency components with a radius greater than $r$ will converge to a small number, suggesting that the high-frequency components (when $r$ is large) are negligible.

2. *The kernel with negligible high-frequency components will weigh the high-frequency components of input images accordingly.*
   This argument can be shown through Convolution Theorem (Bracewell, 1986), which states $\mathbf{w} \circledast \mathbf{x} = \mathcal{F}^{-1}(\mathcal{F}(\mathbf{w}) \odot \mathcal{F}(\mathbf{x}))$, where $\mathcal{F}(\cdot)$ stands for Fourier transform, $\circledast$ stands for convolution operation, and $\odot$ stands for point-wise multiplication. As the theorem states, the convolution operation of images is equivalent to the element-wise multiplication of image frequency domain. Therefore, roughly, if $\mathbf{w}$ has negligible high-frequency components in the frequency domain, it will weigh the high-frequency components of $\mathbf{x}$ accordingly with negligible weights. Naturally, this argument only pertains to a single convolution, and we rely on our intuition that repeated applications of these smooth kernels across multiple convolution layers in a nonlinear deep network will have some cumulative benefit.

Formally, we calculate our regularization term $R_0(\mathbf{w})$ as follows:

$$R_0(\mathbf{w}) = \sum_{i,j} \sum_{h,k \in N(i,j)} (\mathbf{w}_{i,j} - \mathbf{w}_{h,k})^2,$$

We also aim to improve this regularization by trying a few additional heuristics:

- First, we notice that directly appending $R_0(\mathbf{w})$ will sometimes lead to models that achieve the a small value of $R_0(\mathbf{w})$ by directly scaling down the every coefficient of $\mathbf{w}$ proportionally, without changing the fluctuation pattern of the weights. To fix this problem, we directly subtract the scale of $\mathbf{w}$ (*i.e.*, $\sum_{i,j} \mathbf{w}_{i,j}^2$) after $R_0(\mathbf{w})$.

- Another heuristic to fix this same problem is to directly divide $R_0(\mathbf{w})$ by the scale of $\mathbf{w}$. Empirically, we do not observe significant differences between these two heuristics. We settle with the first heuristic because of the difficulty in calculating gradient when a matrix is the denominator.

- Finally, we empirically observe that the regularization above will play a significant role during the early stage of training, but may damage the overall performances later when the regularization pulls towards smoothness too much. To mitigate this problem, we use an exponential function to strengthen the effects of the regularization when the value is big and to weaken it when the value is small.

Overall, our final regularization is:

$$R(\mathbf{w}) = \exp\left( \sum_{i,j} \sum_{h,k \in N(i,j)} (\mathbf{w}_{i,j} - \mathbf{w}_{h,k})^2 - \sum_{i,j} \mathbf{w}_{i,j}^2 \right)$$

In practice, the convolutional kernel is usually a 4-dimensional tensor, while our method only encourages smoothness over the two spatial dimensions corresponding to the 2D images. Thus, we only regularize through these two dimensions broadcasting the operation through the channels.

Because a repeated calculation of each kernel component's distance with its neighbors will double count some pairs, our implementation instead enumerates over all pairs of neighbors, counting each squared difference only once towards the total penalty.

We can directly append the regularization $\lambda R(\mathbf{w})$ to most loss functions, where $\lambda$ is a tuning hyperparameter. In the following experiments, we append $\lambda R(\mathbf{w})$ to the vanilla loss function (cross-entropy loss), Trades loss (Zhang et al., 2019a), adversarial training loss (Madry et al., 2018), and a variation of logit pairing loss (Kannan et al., 2018), as introduced in the following paragraphs.

Adversarial training works by fitting the model using adversarial examples generated on the fly at train time by the threat model. Trades loss fits the model with clean examples while regularizing the softmax of augmented adversarial examples to be close to that produced for corresponding clean examples, a natural alternative is to fit the model with augmented adversarial examples while regularizing the softmax of clean examples to be close to that of the corresponding adversarial examples, which is related to logit pairing. However, to make the comparison consistent, we use a variation of logit pairing, penalizing the KL divergence of softmax (rather than $\ell_2$ distance over logits), following the Trades loss, which also uses KL divergence over softmax as the distance metric.

To be specific, with the standard notations such as $\langle \mathbf{X}, \mathbf{Y} \rangle$ denoting a data set, $\langle \mathbf{x}, \mathbf{y} \rangle$ denoting a sample, the logit pairing loss is formalized as:

$$\min \mathbb{E}_{\langle \mathbf{x}, \mathbf{y} \rangle \sim \langle \mathbf{X}, \mathbf{Y} \rangle} l(f(\mathbf{x}'; \theta); \mathbf{y}) + \gamma k(f_l(\mathbf{x}'; \theta), f_l(\mathbf{x}; \theta))$$
$$\text{where} \quad \mathbf{x}' = \underset{d(\mathbf{x}', \mathbf{x}) \leq \epsilon}{\arg \max} \, l(f(\mathbf{x}'; \theta); \mathbf{y})$$

where $d(\cdot, \cdot)$ and $k(\cdot, \cdot)$ are distance functions, $f_l(\cdot; \cdot)$ denotes the model $f(\cdot; \cdot)$ but outputs the softmax instead of a prediction, $l(\cdot, \cdot)$ is a cost function, $\gamma$ is a tuning hyperparameter, and $\epsilon$ is the upper bound of perturbation. In our following experiments, we consider $d(\cdot, \cdot)$ as $\ell_\infty$ norm following popular adversarial training set-up and $k(\cdot, \cdot)$ as KL divergence following standard Trades loss.

Intuitively, our usage of KL divergence in logit pairing loss is argued to be advantageous because Pinsker's inequality suggests that KL divergence upper-bounds the total variation (TV) distance (*e.g.,* Csiszar & Körner, 2011), the usage of KL divergence can be seen as a regularization that limits the hypothesis space to the parameters that yield small TV distance over perturbations of samples, which is linked to the robustness of an estimator, a topic that has been studied by the statistics community for over decades (*e.g.,* see (Diakonikolas et al., 2019) and references within).

## 4 EXPERIMENTS

To empirically validate our methods, we first consider a simple synthetic experiment to demonstrate the effectiveness of our proposed solutions. Then, with standard data sets such as MNIST (LeCun, 1998), FashionMNIST (Xiao et al., 2017), CIFAR10 (Krizhevsky & Hinton, 2009) and Restricted ImageNet (Tsipras et al., 2019), we evaluate our methods with well-established criteria, such as $\ell_\infty$ bounded accuracy. We also leverage saliency-based visualization methods to understand how the model understands each class. Most experiments are conducted with a simple basic convolutional neural network with two convolution layers and two fully connected layers, while the CIFAR10 experiment is conducted with ResNet18 and Restricted ImageNet experiment is conducted with ResNet50 (more details of the models are in the Appendix). As we mentioned previously, we apply the new regularization to four different losses: the vanilla loss (denoted as **V**), Trades loss (denoted as **T**) (Zhang et al., 2019a), adversarial training (denoted as **A**) (Madry et al., 2018), and our variation of logit pairing (denoted as **L**). **T**, **A**, **L** all adopt $\ell_\infty$ norm bounded PGD as the threat model. We use **VR**, **TR**, **AR**, **LR** to denote the methods after our regularization is plugged in. We evaluate our methods against a wide range of adversarial attack methods, including FGSM (Goodfellow et al., 2015), PGD (Madry et al., 2018), C&W (Carlini & Wagner, 2017), DeepFool (both $\ell_2$ and $\ell_\infty$) (Moosavi-Dezfooli et al., 2016), ADef, a method that iteratively applies small deformations to the image (Alaifari et al., 2019) and Salt&Pepper, a black-box method that adds noise to the image. For these attack methods, we use the default parameters in Foolbox (Rauber et al., 2017), and our experiments suggest that these default parameters are effective enough in most cases. For every data set, we first tune the $\ell_\infty$ norm perturbation bound of adversarial training method and then use the same setting for Trades loss and variation of logit pairing. We tune $\gamma$ within $\{0.1, 1.0, 10.0\}$ and tune $\lambda$ within $\{0.01, 0.1, 1.0, 10.0, 100.0\}$.

### 4.1 SYNTHETIC EXPERIMENTS FOR SANITY CHECKING

We first use a basic data set of four shapes[1] to test whether our proposed method helps regularize the model to behave as we desire. Each image in this data set has a white background and a black foreground depicting one of the four shapes: circle, star, square, and triangle. Our goal is to train a convolutional neural network to classify the images into one of these four shapes.

We compare the models trained the four basic losses **V**, **T**, **A**, **L** and these models with our regularization, denoted as **VR**, **TR**, **AR**, and **LR**, when $\lambda = 100.0$. To further test our idea, we also test the regularization with the hyperparameter set to a negative value $\lambda = -100.0$ to inspect the consequences when we regularize the model towards high-frequency kernels. Resulting models are denoted as **VH**, **TH**, **AH**, **LH** respectively according to the basic losses.

We report our inspections in Figure 1: Figure 1(a) visualizes of the convolution kernel (due to the limitation of space, we only visualize the first four convolutional kernels); Figure 1(b) visualizes the corresponding frequency domain in absolute values; Figure 1(c) visualizes the internal representation after an image depicting *star* is passed through the kernels.

Figure 1 (a) shows that our regularization guides the model towards a smooth kernel, across all the basic losses. Also, if we apply our regularization with a negative parameter, then the weights of the resulting kernel tend to fluctuate more dramatically. Figure 1 (b) validates our argument that a smooth kernel only has negligible high-frequency components. As we can see, the frequency domain corresponding to the kernels when our regularization is applied shows significant differences in low-frequency components (center of the visualization) and high-frequency components (periphery of the visualization). Figure 1 (c) further validates our intuition, showing that in comparison to

---

[1]https://www.kaggle.com/smeschke/four-shapes

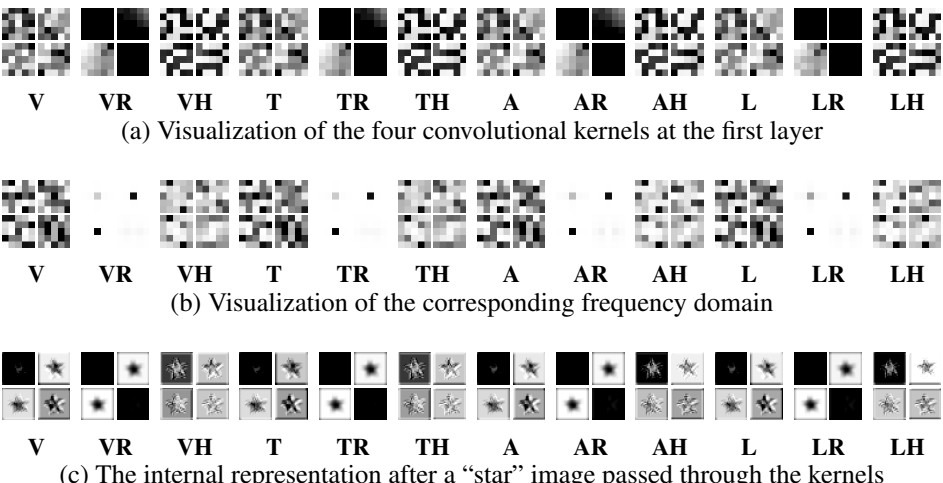

(a) Visualization of the four convolutional kernels at the first layer

**V    VR    VH    T    TR    TH    A    AR    AH    L    LR    LH**

(b) Visualization of the corresponding frequency domain

**V    VR    VH    T    TR    TH    A    AR    AH    L    LR    LH**

(c) The internal representation after a "star" image passed through the kernels

**V    VR    VH    T    TR    TH    A    AR    AH    L    LR    LH**

Figure 1: Visualization of the first four convolutional kernels of each model (first row), corresponding frequency domain (second row), and the internal representation when an image is passed into the kernel (third row).

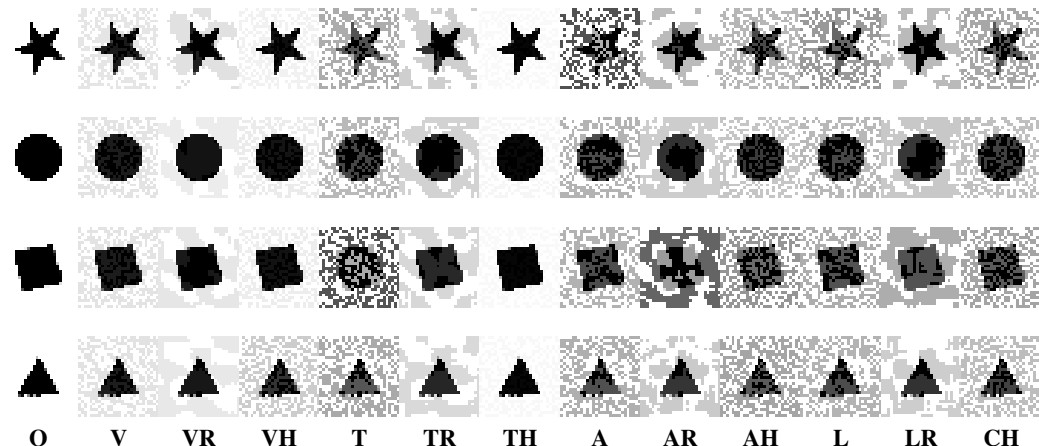

**O    V    VR    VH    T    TR    TH    A    AR    AH    L    LR    CH**

Figure 2: The averaged most deceptive images (one for each class) generated by FGSM for the testing models. **O** stands for the original example.

internal representations summarized by kernels from basic losses, those influenced by our regularization are more sensitive to the low-frequency signal (e.g. the shape of the input), and the internal representation with our regularization when the parameter is negative tends to focus more on the high-frequency signals.

Further, we check the mechanism of our model by inspecting how adversarial examples deceive the models. Figure 2 shows the four on-average most deceptive adversarial examples (the models predict incorrectly with the highest confidence) generated by FGSM. Notations follow the same convention as the previous case, and **O** denotes the original image.

While many images have to be perturbed with a human perceivable amount to deceive the model, we can notice that the adversarial examples for models with our regularization (⋆**R**) tend to behave in way that can be understood by a human. The most convincing examples are at the first row for **A** and **AR**, where we can clearly see that the adversarial examples alter the decisions from *star* to *circle*. Other adversarial examples for ⋆**R** models also introduce large areas that shall be interpreted as the shape. In contrast, adversarial examples for other models tend to introduce scattered patches, which

Table 1: Prediction accuracy over adversarial examples. For MNIST and FashionMNIST, we do not limit the $\epsilon$ of adversrial examples. For CIFAR10 and Restricted ImageNet, we limit the $\epsilon$ to be 0.1 and 0.05 respectively ($\ell_\infty$ difference, the maximum is 1.0). Best performance for each data set for each attack method is highlighted (except when the best performance is below random guess).

| | Model | clean | FGSM | PGD | DF($\ell_2$) | DF($\ell_\infty$) | C&W | ADef | S&P |
|---|---|---|---|---|---|---|---|---|---|
| MNIST | **V** | **0.989** | 0.013 | 0 | 0 | 0 | 0 | 0 | 0 |
| | **VR** | 0.986 | 0.05 | 0 | **0.969** | **0.821** | 0 | 0.01 | 0 |
| | **T** | **0.989** | 0.01 | 0.039 | 0.404 | 0.344 | 0.033 | 0.131 | 0.001 |
| | **TR** | 0.983 | 0.138 | **0.53** | 0.547 | 0.465 | 0.162 | **0.772** | **0.109** |
| | **A** | **0.989** | 0.075 | 0.021 | 0.349 | 0.094 | 0.129 | 0.071 | 0 |
| | **AR** | 0.988 | 0.223 | 0.017 | 0.314 | 0.088 | 0.163 | 0.044 | 0 |
| | **L** | 0.951 | 0.136 | 0.022 | 0.466 | 0.072 | 0.094 | 0.12 | 0.001 |
| | **LR** | 0.952 | **0.292** | 0.012 | 0.934 | 0.662 | **0.295** | 0.204 | 0.001 |
| Fashion MNIST | **V** | 0.908 | 0.012 | 0.001 | 0.001 | 0.001 | 0 | 0 | 0 |
| | **VR** | 0.885 | 0.009 | 0.075 | 0.758 | 0.635 | 0.005 | 0.413 | 0.002 |
| | **T** | **0.918** | 0.04 | 0.151 | 0.275 | 0.271 | 0.014 | 0.157 | 0.003 |
| | **TR** | 0.873 | 0.055 | **0.166** | **0.785** | **0.741** | 0.086 | **0.554** | 0.003 |
| | **A** | 0.869 | 0.065 | 0.038 | 0.141 | 0.101 | 0.015 | 0.081 | 0.035 |
| | **AR** | 0.747 | 0.06 | 0.072 | 0.231 | 0.225 | 0.121 | 0.218 | 0.002 |
| | **L** | 0.86 | 0.082 | 0.034 | 0.216 | 0.044 | 0.031 | 0.089 | 0.034 |
| | **LR** | 0.752 | **0.176** | 0.103 | 0.573 | 0.448 | **0.354** | 0.335 | 0.012 |
| CIFAR10 ($\epsilon = 0.1$) | **V** | **0.85** | 0.02 | 0.02 | 0.02 | 0 | 0 | 0.64 | **0.85** |
| | **VR** | 0.82 | 0.03 | 0.02 | 0.02 | 0 | 0.01 | 0.64 | 0.82 |
| | **T** | 0.78 | 0.01 | 0.01 | 0.01 | 0 | 0 | 0.62 | 0.77 |
| | **TR** | 0.81 | 0.03 | 0.03 | 0.03 | 0.01 | 0 | **0.74** | 0.81 |
| | **A** | 0.72 | 0.07 | 0.55 | 0.55 | 0.48 | 0.03 | 0.68 | 0.72 |
| | **AR** | 0.72 | 0.09 | **0.58** | **0.58** | **0.57** | 0.02 | 0.72 | 0.72 |
| | **L** | 0.66 | **0.12** | 0.56 | 0.56 | 0.55 | 0.03 | 0.65 | 0.66 |
| | **LR** | 0.68 | 0.11 | 0.57 | 0.57 | 0.54 | 0.03 | 0.62 | 0.68 |
| Restricted ImageNet ($\epsilon = 0.05$) | **V** | 0.851 | 0.063 | 0 | 0.017 | 0 | 0.004 | 0.851 | 0.851 |
| | **VR** | **0.855** | 0.047 | 0.001 | 0.025 | 0.001 | 0.006 | **0.855** | **0.855** |
| | **A** | 0.404 | 0.081 | 0.022 | 0.33 | 0.03 | 0.189 | 0.404 | 0.404 |
| | **AR** | 0.637 | 0.078 | 0.021 | 0.476 | 0.023 | 0.239 | 0.637 | 0.637 |
| | **L** | 0.587 | 0.11 | 0.021 | **0.477** | 0.034 | **0.257** | 0.587 | 0.587 |
| | **LR** | 0.591 | 0.096 | 0.02 | 0.468 | 0.036 | 0.254 | 0.591 | 0.591 |

will probably not be considered as the shape for most people. Also, if we apply our regularization with a negative parameter (⋆**H**), the patches tend to behave in a more shattered manner.

## 4.2 STANDARD NUMERICAL EVALUATION

In Table 1, we report the prediction accuracy over the generated adversarial examples across the attack methods. For MNIST and FashionMNIST, we do not limit the $\epsilon$ of adversarial examples. In principle, when there is no $\epsilon$ constraint, one should always be able to find the adversarial example for any sample, however, in practice, many search attempts fail when the attack methods are set with the default hyperparameters in Foolbox. We consider these failures of searches (under default parameters) also a measure of the robustness of models. For CIFAR10 and Restricted ImageNet, we set the $\epsilon$ to be 0.1 and 0.05, respectively (the maximum pixel value is 1.0).

Overall, across most of the settings, our regularization helps achieve numerically the best adversarially robust models. Impressively, for MNIST and FashionMNIST, for some attack methods (*e.g.*, both versions of DeepFool), our regularization can improve the robustness significantly even when only applied to the vanilla training loss, suggesting the importance of the smooth regularization. Also, for these two datasets, the improvements of our regularization are mostly significant over the non-regularized counterparts. For CIFAR10 and Restricted ImageNet, the performance gains are less significant but still observable. In the Appendix, we report the accuracy and $\epsilon$ curves over $\ell_0$,

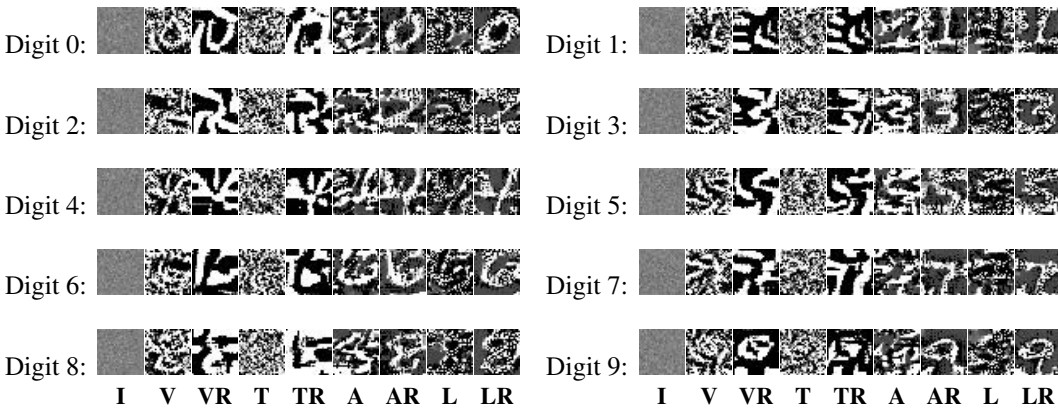

Figure 3: Sample-independent interpretation of models trained over MNIST. **I** stands for the input.

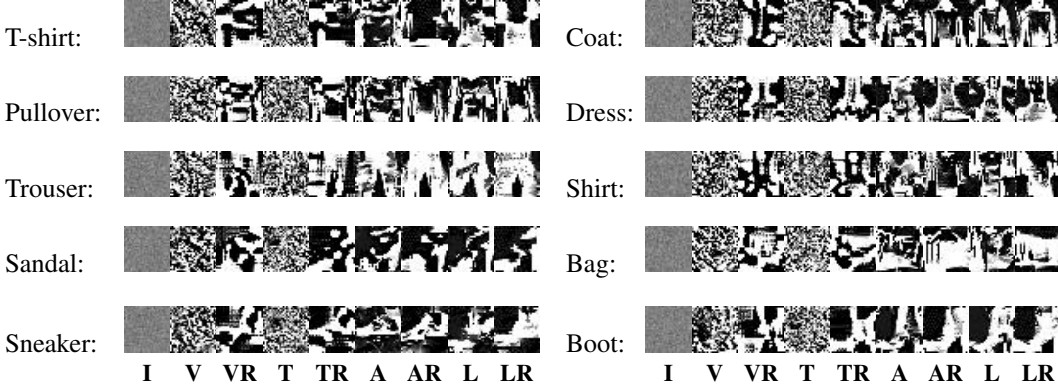

Figure 4: Sample-independent interpretation of models for FashionMNIST. **I** stands for the input.

$\ell_2$, and $\ell_\infty$ distances, for more thorough comparisons. In general, the performances evaluated by the curves are consistent with results in Table 1.

### 4.3 INSPECTING MODEL'S PERCEPTION OF CLASS

We also leverage one of the most classic model-interpretation methods, activation maximization (Erhan et al., 2009), to further demonstrate the strength of our regularization. Concretely, we follow (Simonyan et al., 2013; Engstrom et al., 2019) to maximize the logit of a certain class so that the most representative features of that class can be exaggerated in the input image. Specifically, with an input image of Gaussian noise, we apply projected gradient descent 10000 iterations with learning rate 0.001 to update the input image. Notice that the interpretation is sample-independent.

Figure 3 depicts what the models consider to be the digits. While **V** and **T** can barely be interpreted to human, when our regularization is plugged in, the patterns appear to be observable, with impressive examples such as Digit 0, 2, 3. **A** can also deliver interpretable decisions (*e.g.*, Digit 3 and 5), and our regularization significantly helps in other cases, such as Digit 0, 1, 2, 4, and 8. Figure 4 shows a similar story for FashionMNIST dataset: while **A** might have the cleanest interpretation for the "sneaker" case, our regularization (especially **AR**) probably has the best interpretation in all other cases, with good examples such as "Trouser," "Dress," and "Boot." Interestingly, **AR** is the only method that interprets "Bag" with a strap, and the average image of all training "Bag" samples in FashionMNIST is a bag with a strap. Figure 5 shows the visualization of models trained on CIFAR10. While **A** seems to have the best interpretation in "horse" case, **AR** and **LR** have equal or better interpretation in comparison with **A** in other cases. Impressively, only **AR** and **LR** understand "bird," and only **AR** understands "deer". Figure 6 shows the visualization for Restricted ImageNet

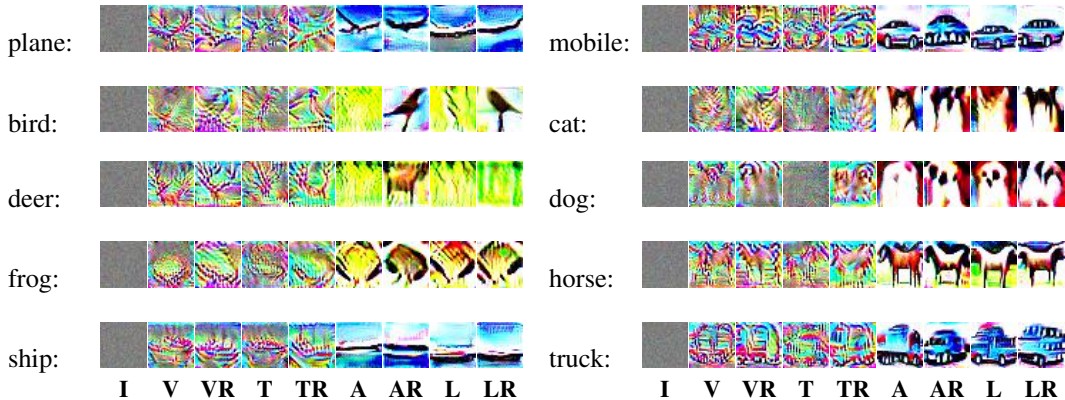

Figure 5: Sample-independent interpretation of models trained over CIFAR10. **I** stands for the input.

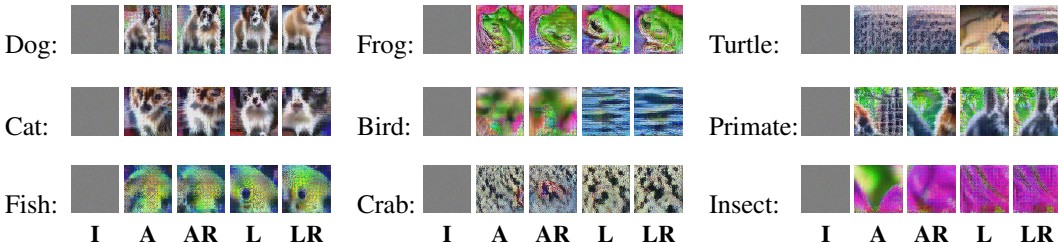

Figure 6: Sample-independent interpretation of models trained over Restricted ImageNet. **I** stands for the input. Results of simpler models are not shown because they cannot be interpreted well.

(results of simpler models are not shown because they cannot be interpreted). **AR** is the only method that can describe the outline of the "bird" and "crab" classes, while these models seem to remain more or less the similar interpretation power for other labels.

Other results, such as visualization of targeted attack through saliency-based methods and selective visualization of adversarial examples generated along with the experiments, are shown in the Appendix. Overall, the empirical evidence supports our intuition in Section 3: the regularization helps push the model to focus on the low-frequency components of the image and thus leads to more perceptually-aligned gradients.

## 5 CONCLUSION

Inspired by neuroscience literature emphasizing the connection between low-frequency components and shape recognition (Bar, 2004; Awasthi et al., 2011), we proposed a smooth kernel regularization that forces the CNN to learn smooth convolutional kernels (kernels with small differences among adjacent weights) during training. As the relation between smoothness and low-frequency can be argued intuitively and supported by some known results in proved theorems (Titchmarsh, 1948; Bracewell, 1986; Platonov, 2005), our regularization should help the model to depend more on the low-frequency components of images. To verify the effectiveness of the regularization, we plug in the idea onto multiple training losses, including the vanilla loss, Trades loss (Zhang et al., 2019a), the adversarial training loss (Madry et al., 2018), as well as a variation of Logit Pairing loss (Kannan et al., 2018). With seven different attack methods, we demonstrate the empirical strength of our regularization with standard numerical evaluations. Further, we also leverage the standard model interpretation methods to explain the decision of models, showing that our technique, like those demonstrated by Santurkar et al. (2019), tends to result in more perceptually-aligned gradients.

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

## A    MODEL AND HYPERPARAMETER CHOICES

For MNIST and FashionMNIST data set, the model is a simple two-layer architecture with two convolutional layers and two fully connected layers. The $\ell_\infty$ perturbation bound of PGD is set to 0.3/1.0 for MNIST and 0.1/1.0 for FashionMNIST. For CIFAR10, the model is a ResNet18, and the $\ell_\infty$ perturbation bound of PGD is set to 0.03/1.0 (roughly 8/255). For Restricted ImageNet, the model is a ResNet50, and the $\ell_\infty$ perturbation bound of PGD is set to 0.005/1.0, then along the processing process, the pixel values of the images are divided by the standard deviation (0.2575), so is the perturbation bound. Also, for Restricted ImageNet, we continue with either the standard ImageNet-pretrained ResNet50 (for **V** and **T** losses) or the adversarially trained ResNet50 on Restricted ImageNet (Santurkar et al., 2019) (for **A** and **L** losses). With our hardware settings (NVIDIA 1080Ti), we cannot effectively train Trades loss over ResNet50.

# B ACCURACY-EPSILON CURVES

The accuracy-epsilon curves are shown for $\ell_0$, $\ell_2$ and $\ell_\infty$ bounds are shown in Figure 7, Figure 8, and Figure 9.

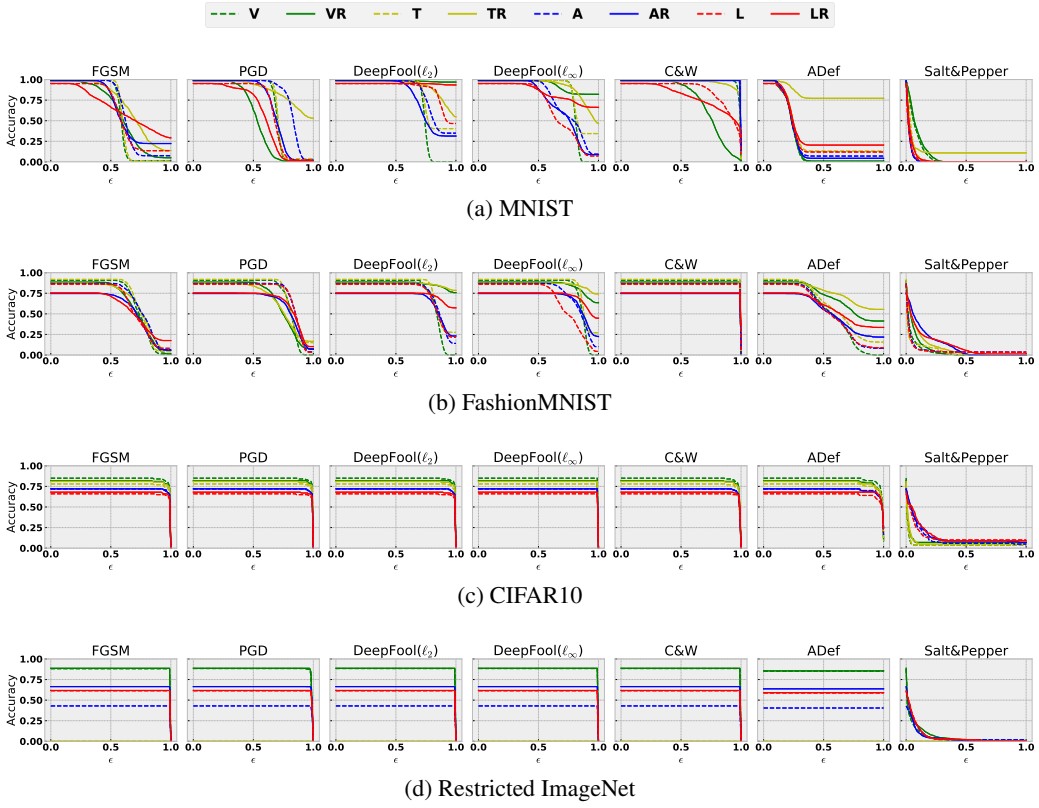

Figure 7: Accuracy over $\ell_0$ bound across different attack methods with models trained with the three data sets. Novel methods are denoted with solid lines.

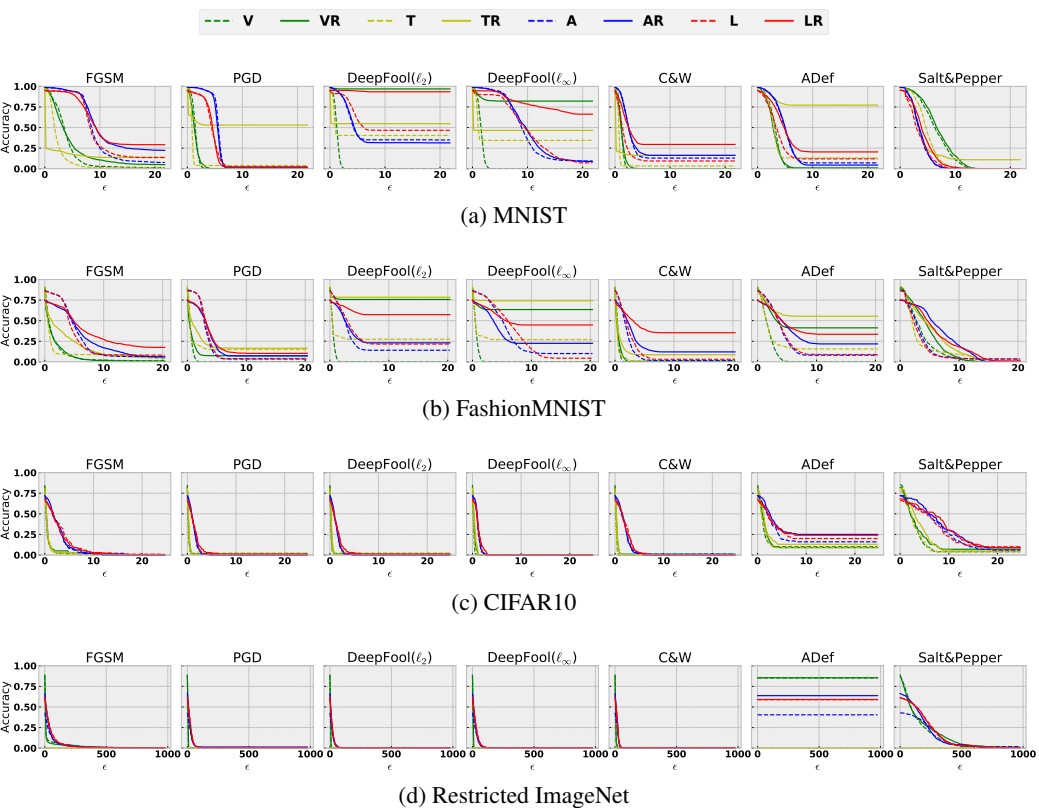

Figure 8: Accuracy over $\ell_2$ bound across different attack methods with models trained with the three data sets. Novel methods are denoted with solid lines.

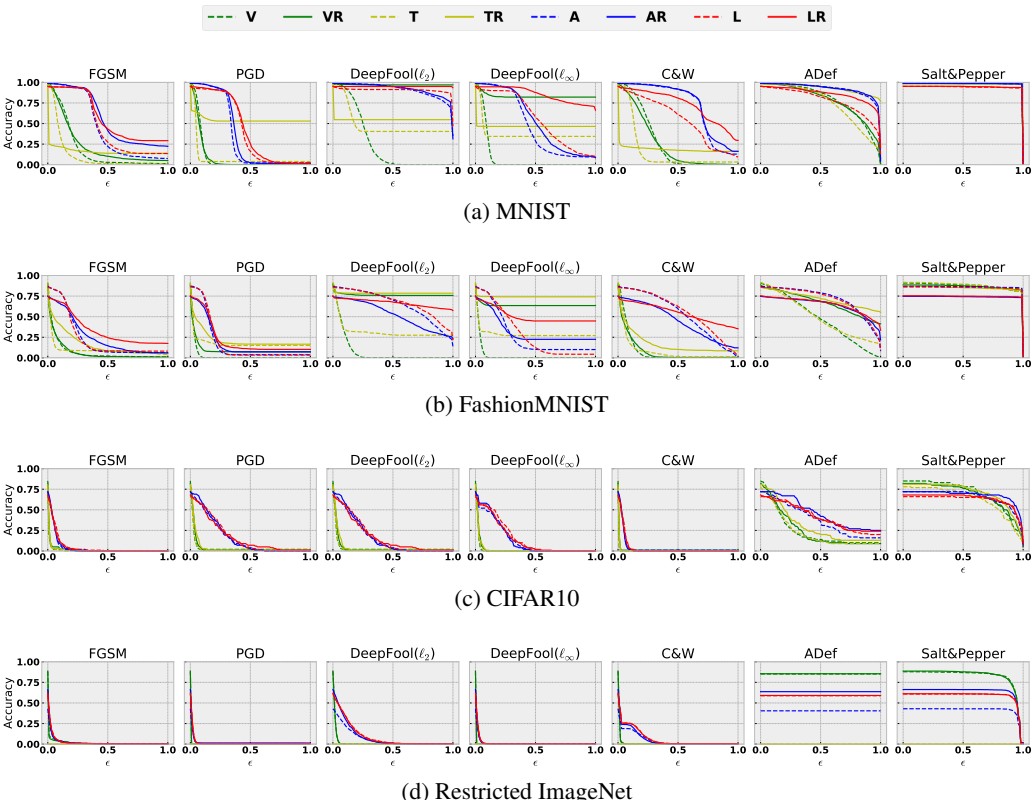

Figure 9: Accuracy over $\ell_\infty$ bound across different attack methods with models trained with the three data sets. Novel methods are denoted with solid lines.

## C TARGETED ATTACK

We also take advantage of the gradient to perform targeted attack, as shown in following figures. The titles of the columns describe the original class, and the titles of the rows describe the target classes.

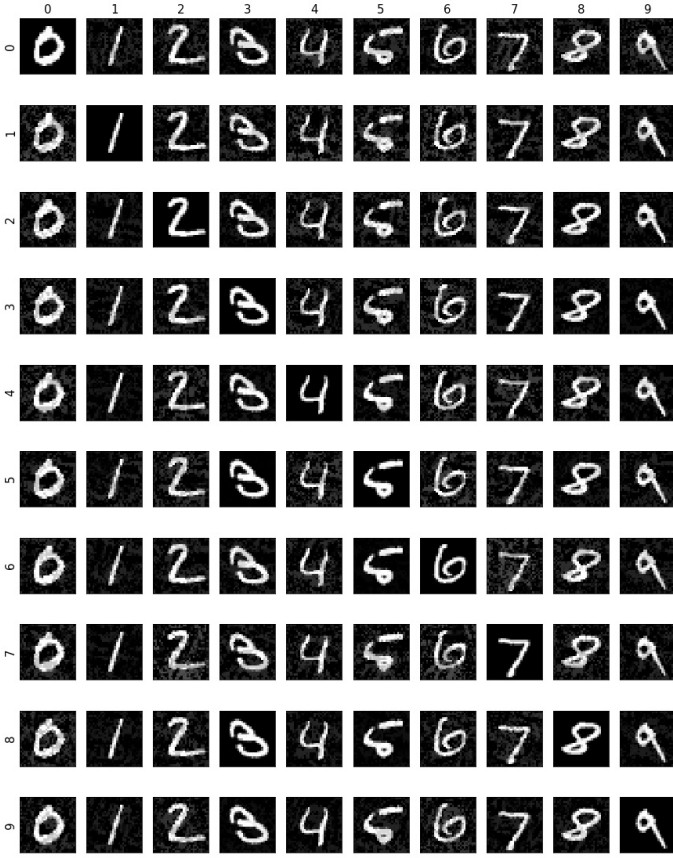

Figure 10: targeted attack visualization for **V** trained with MNIST. The titles of the columns describe the original class, and the titles of the rows describe the target classes.

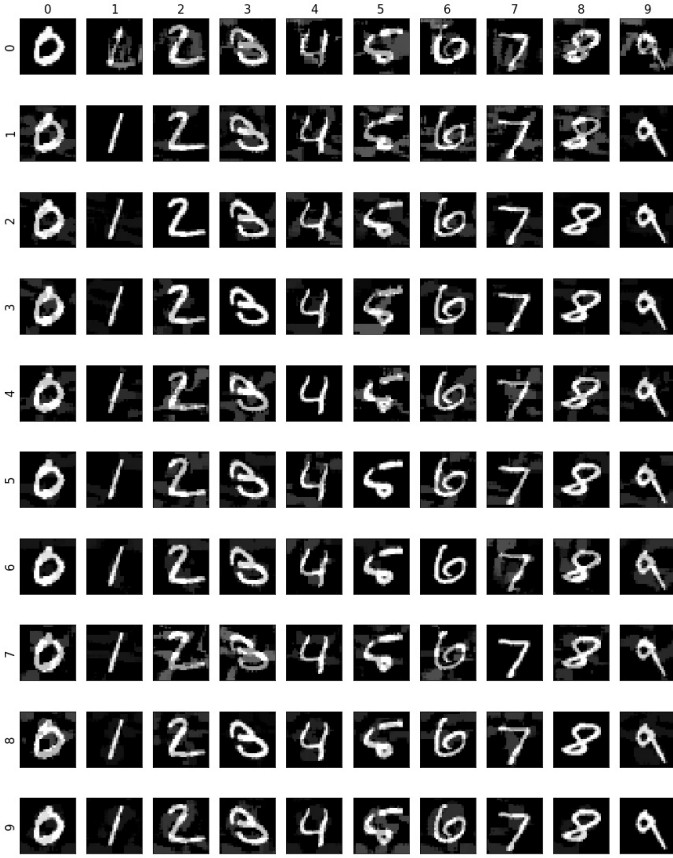

Figure 11: targeted attack visualization for **VR** trained with MNIST. The titles of the columns describe the original class, and the titles of the rows describe the target classes.

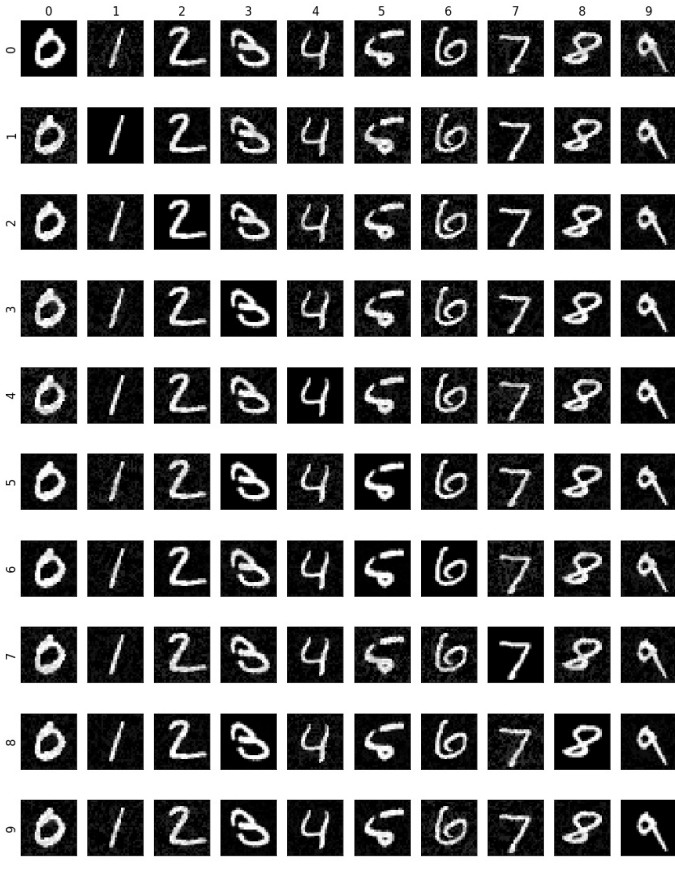

Figure 12: targeted attack visualization for **T** trained with MNIST. The titles of the columns describe the original class, and the titles of the rows describe the target classes.

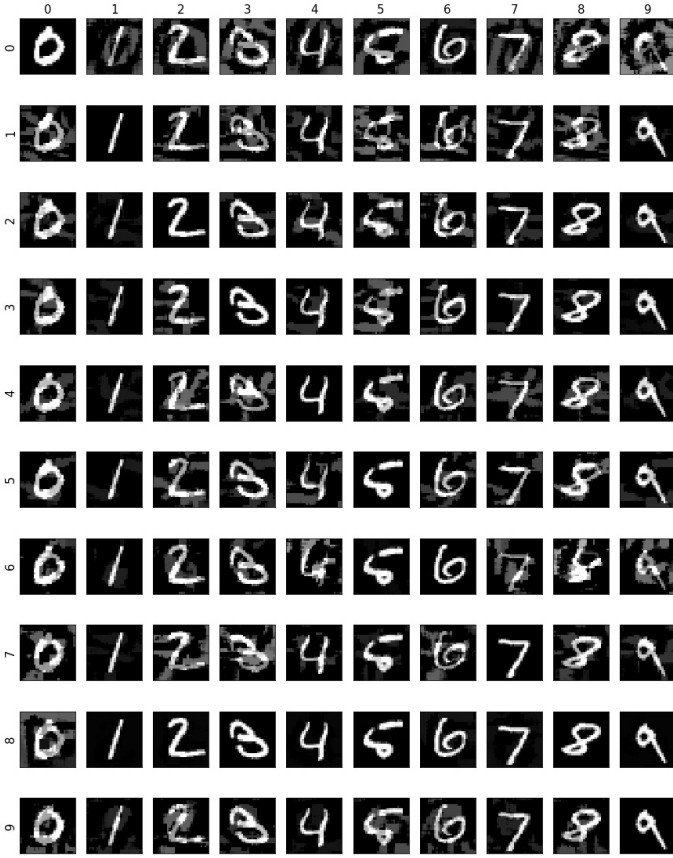

Figure 13: targeted attack visualization for **TR** trained with MNIST. The titles of the columns describe the original class, and the titles of the rows describe the target classes.

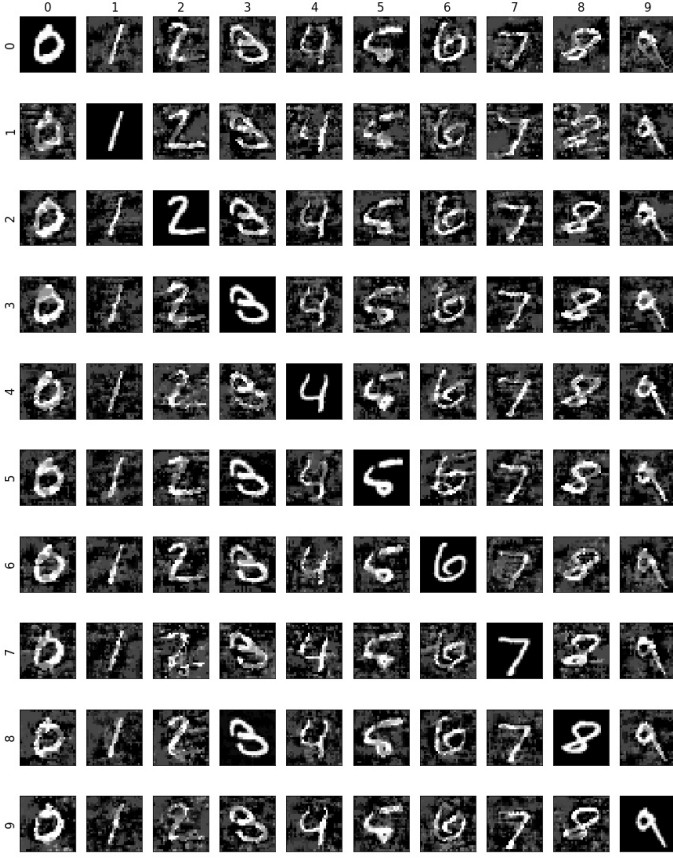

Figure 14: targeted attack visualization for **A** trained with MNIST. The titles of the columns describe the original class, and the titles of the rows describe the target classes.

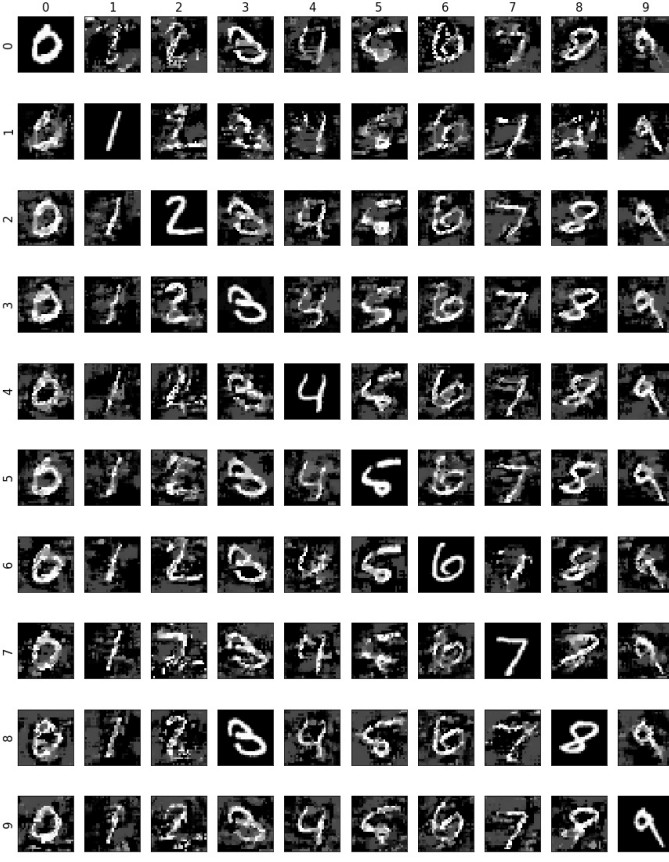

Figure 15: targeted attack visualization for **AR** trained with MNIST. The titles of the columns describe the original class, and the titles of the rows describe the target classes.

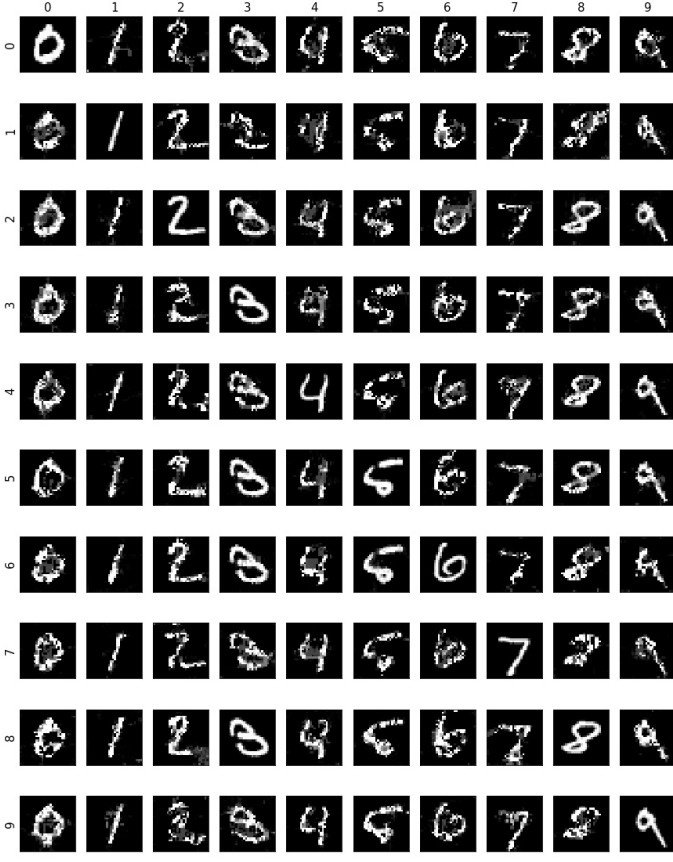

Figure 16: targeted attack visualization for **L** trained with MNIST. The titles of the columns describe the original class, and the titles of the rows describe the target classes.

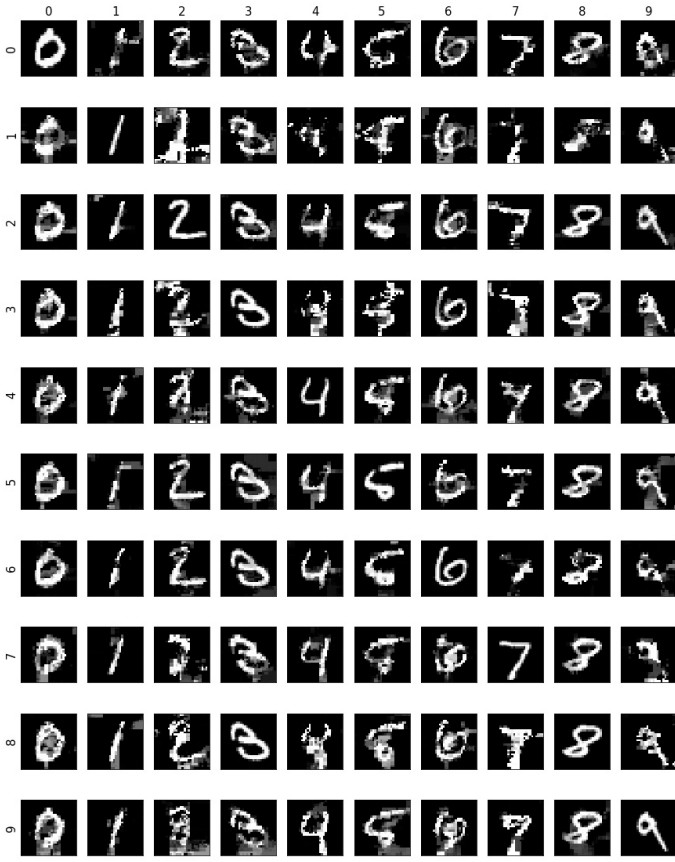

Figure 17: targeted attack visualization for **LR** trained with MNIST. The titles of the columns describe the original class, and the titles of the rows describe the target classes.

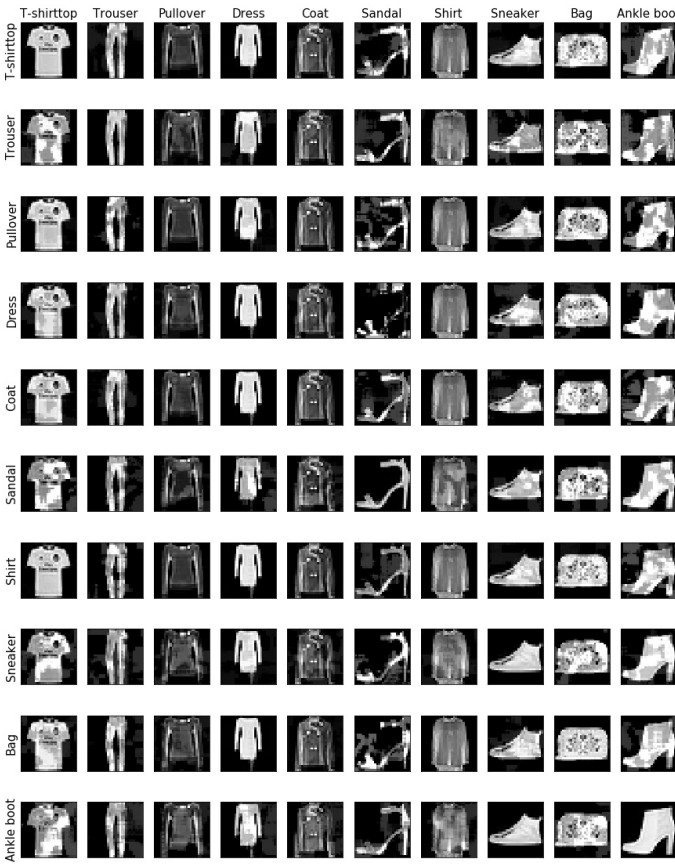

Figure 18: targeted attack visualization for **V** trained with FashionMNIST. The titles of the columns describe the original class, and the titles of the rows describe the target classes.

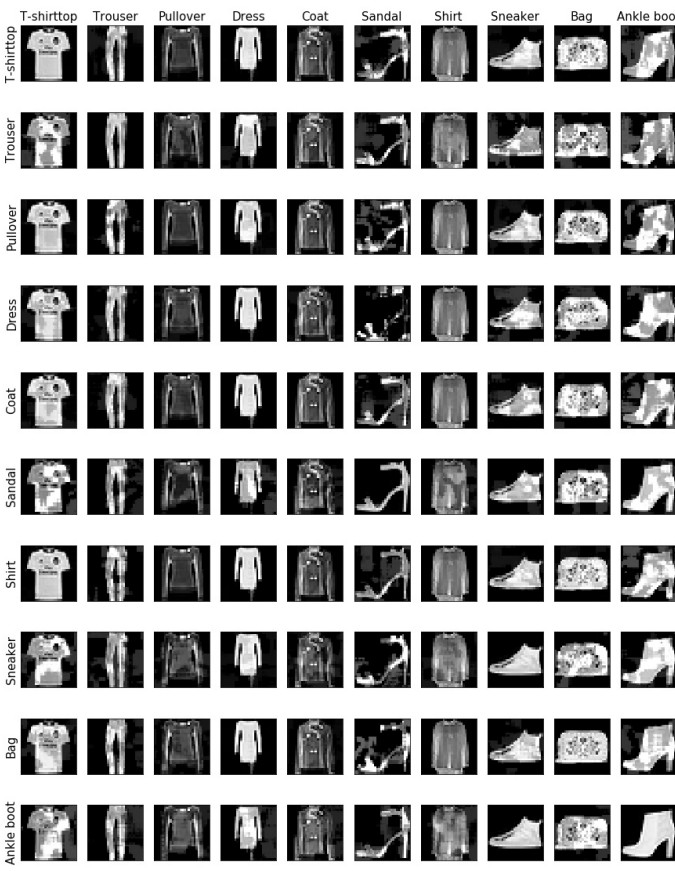

Figure 19: targeted attack visualization for **VR** trained with FashionMNIST. The titles of the columns describe the original class, and the titles of the rows describe the target classes.

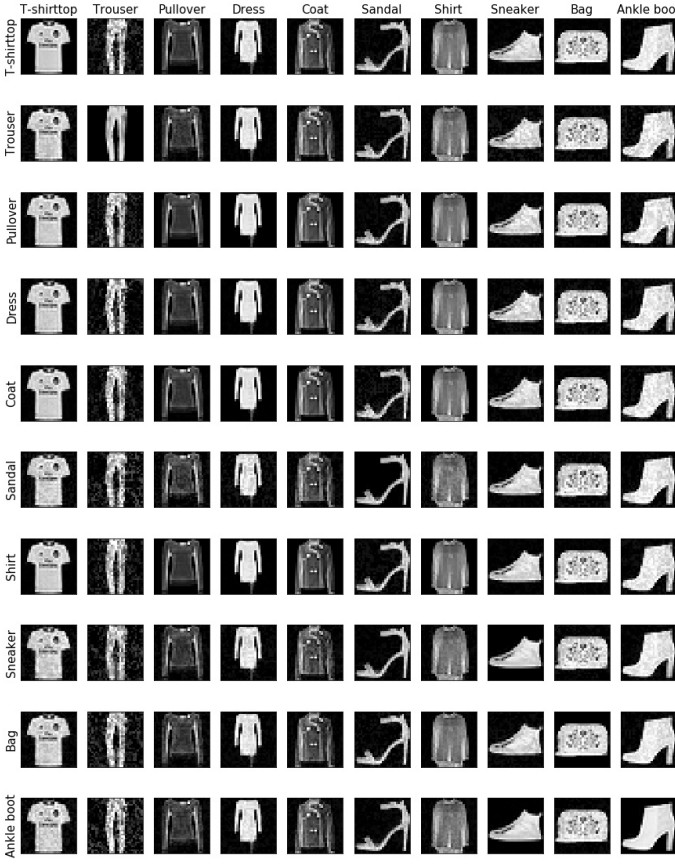

Figure 20: targeted attack visualization for **T** trained with FashionMNIST. The titles of the columns describe the original class, and the titles of the rows describe the target classes.

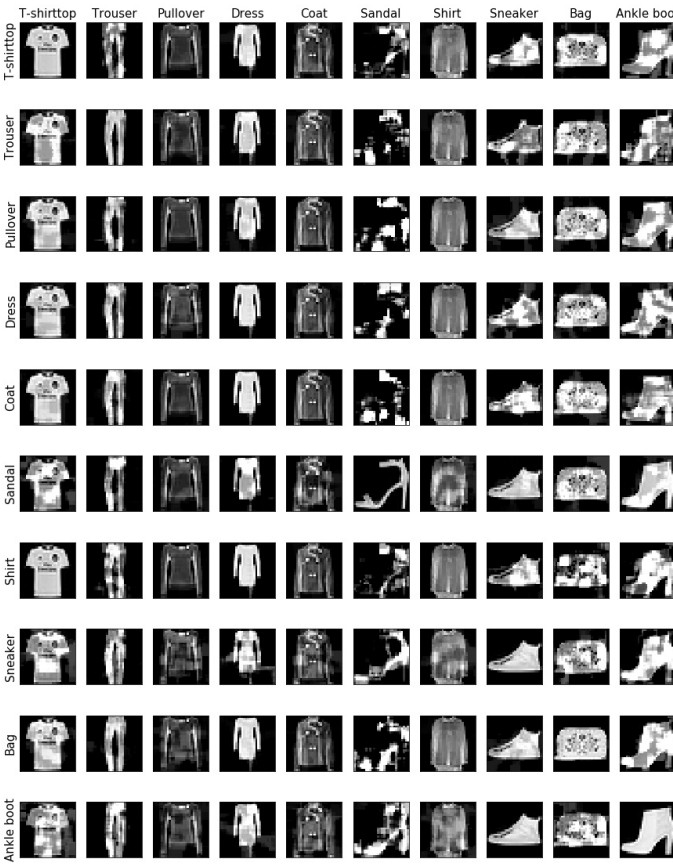

Figure 21: targeted attack visualization for **TR** trained with FashionMNIST. The titles of the columns describe the original class, and the titles of the rows describe the target classes.

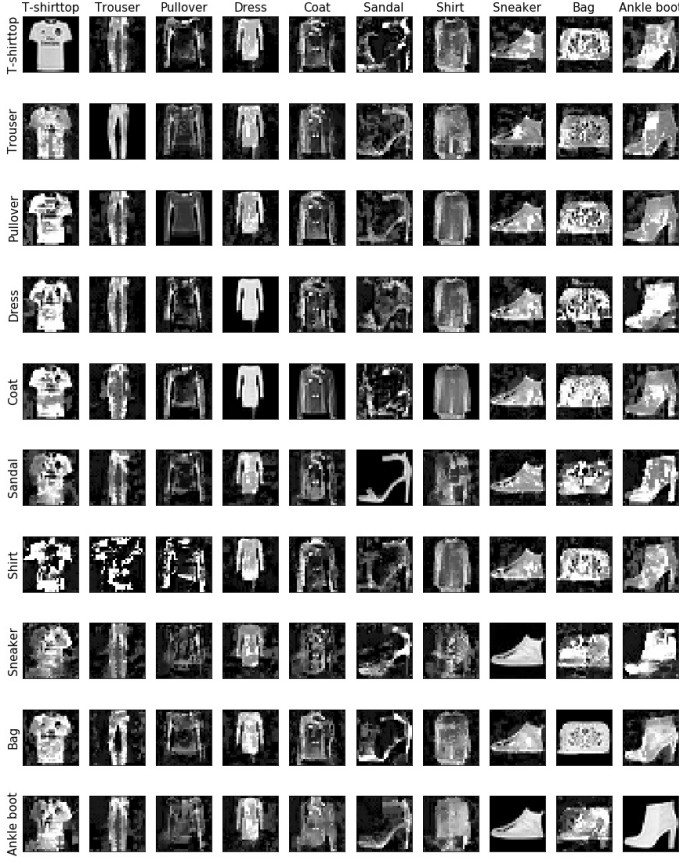

Figure 22: targeted attack visualization for **A** trained with FashionMNIST. The titles of the columns describe the original class, and the titles of the rows describe the target classes.

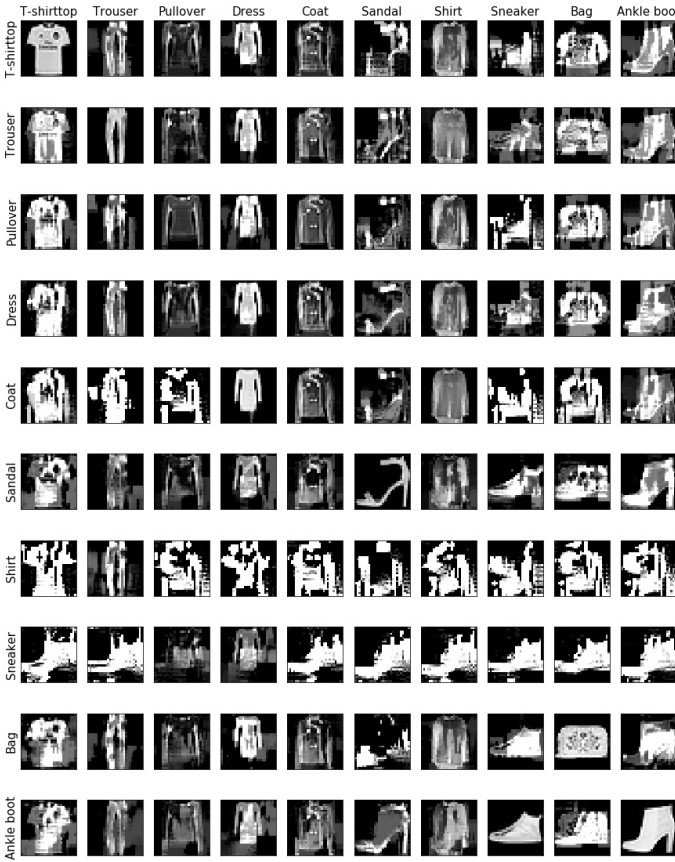

Figure 23: targeted attack visualization for **AR** trained with FashionMNIST. The titles of the columns describe the original class, and the titles of the rows describe the target classes.

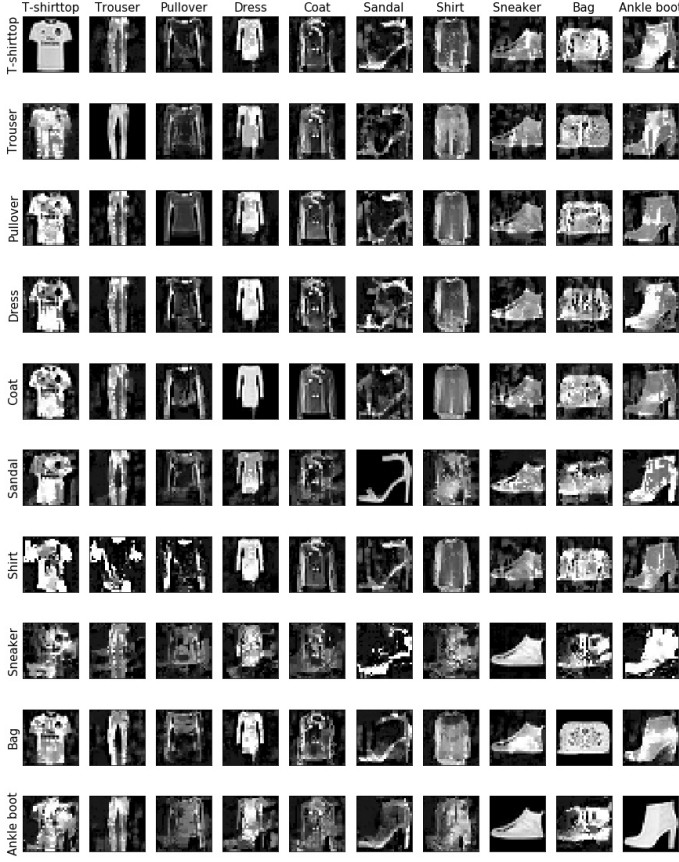

Figure 24: targeted attack visualization for **L** trained with FashionMNIST. The titles of the columns describe the original class, and the titles of the rows describe the target classes.

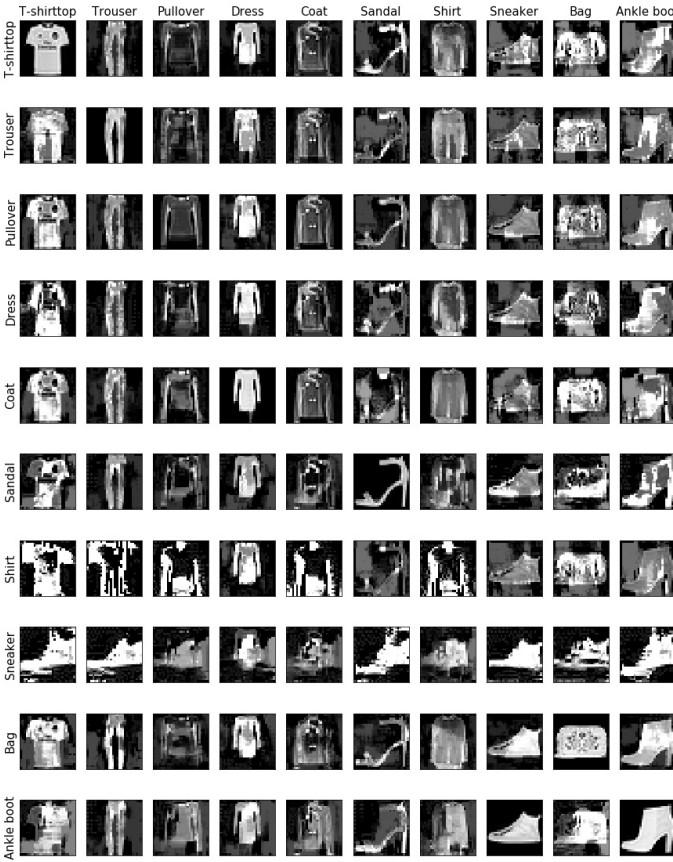

Figure 25: targeted attack visualization for **LR** trained with FashionMNIST. The titles of the columns describe the original class, and the titles of the rows describe the target classes.

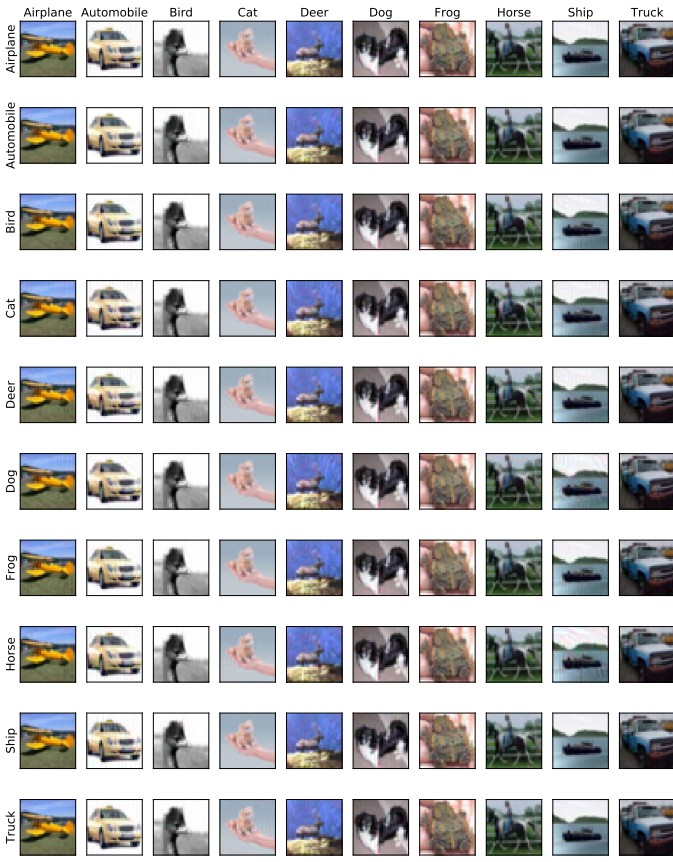

Figure 26: targeted attack visualization for **V** trained with CIFAR10. The titles of the columns describe the original class, and the titles of the rows describe the target classes.

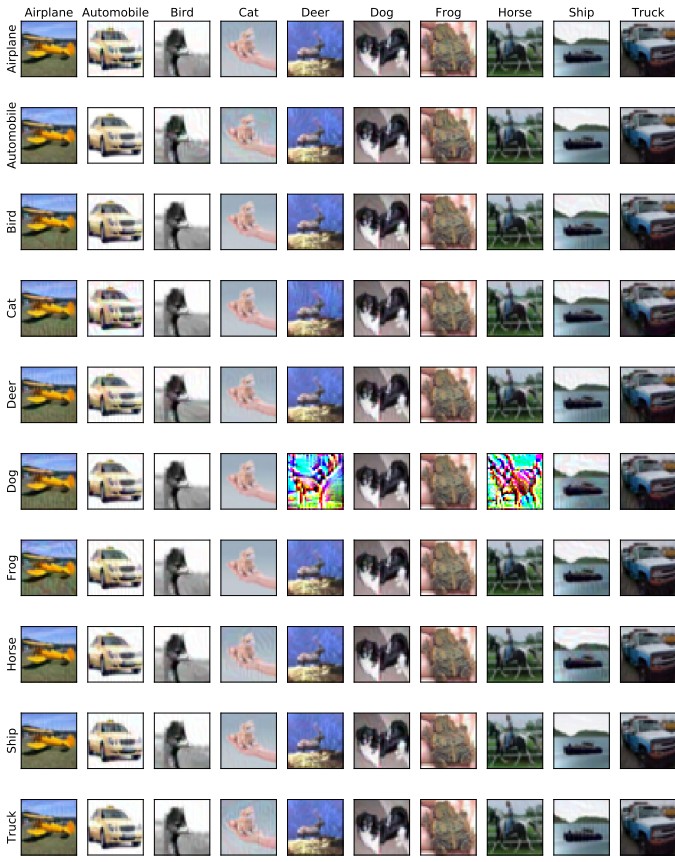

Figure 27: targeted attack visualization for **VR** trained with CIFAR10. The titles of the columns describe the original class, and the titles of the rows describe the target classes.

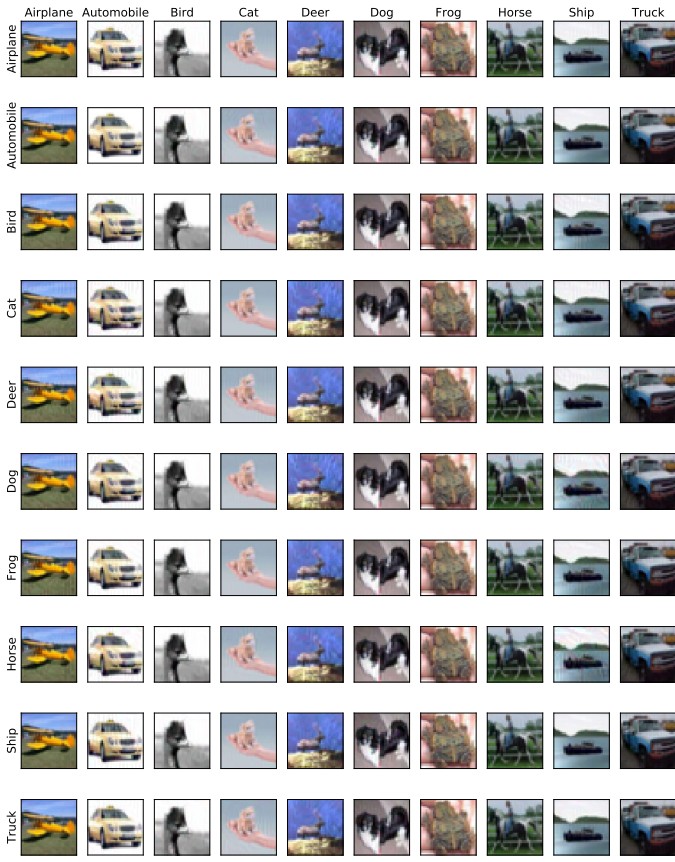

Figure 28: targeted attack visualization for **T** trained with CIFAR10. The titles of the columns describe the original class, and the titles of the rows describe the target classes.

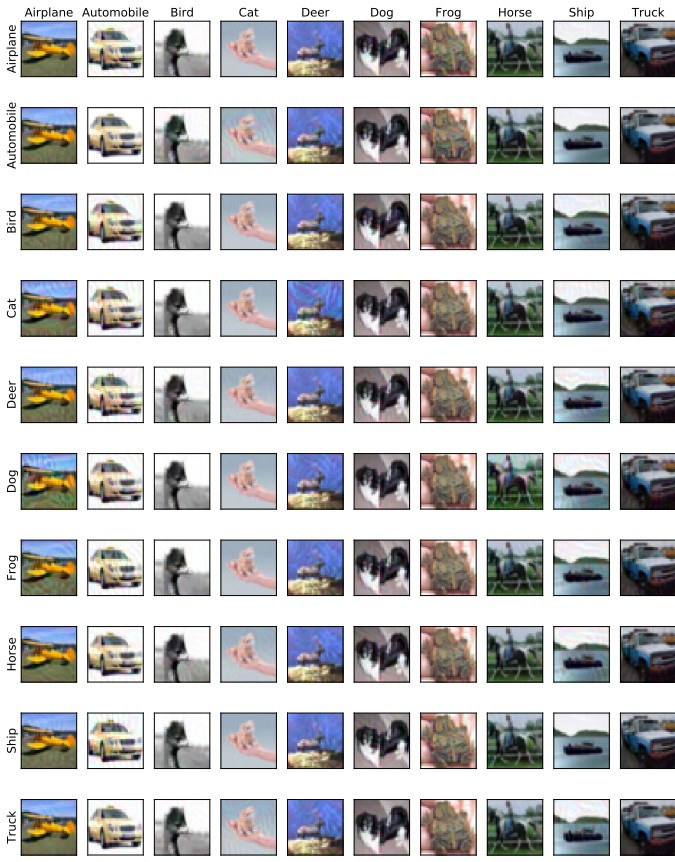

Figure 29: targeted attack visualization for **TR** trained with CIFAR10. The titles of the columns describe the original class, and the titles of the rows describe the target classes.

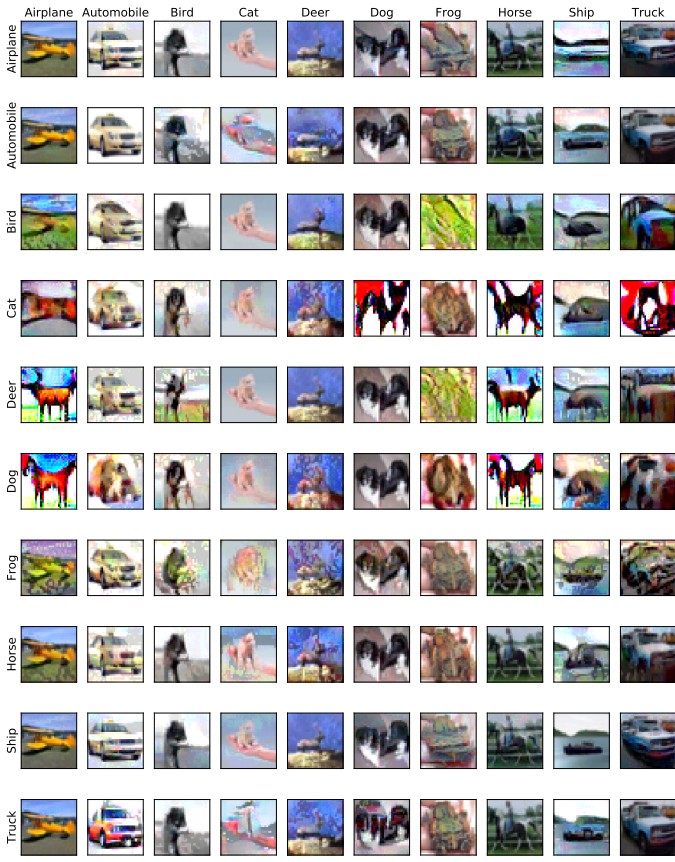

Figure 30: targeted attack visualization for **A** trained with CIFAR10. The titles of the columns describe the original class, and the titles of the rows describe the target classes.

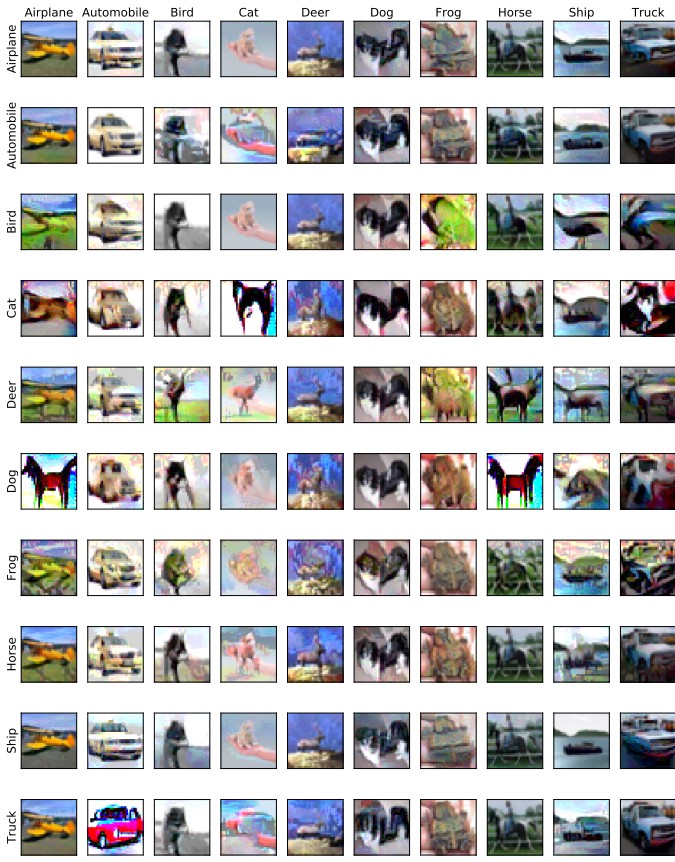

Figure 31: targeted attack visualization for **AR** trained with CIFAR10. The titles of the columns describe the original class, and the titles of the rows describe the target classes.

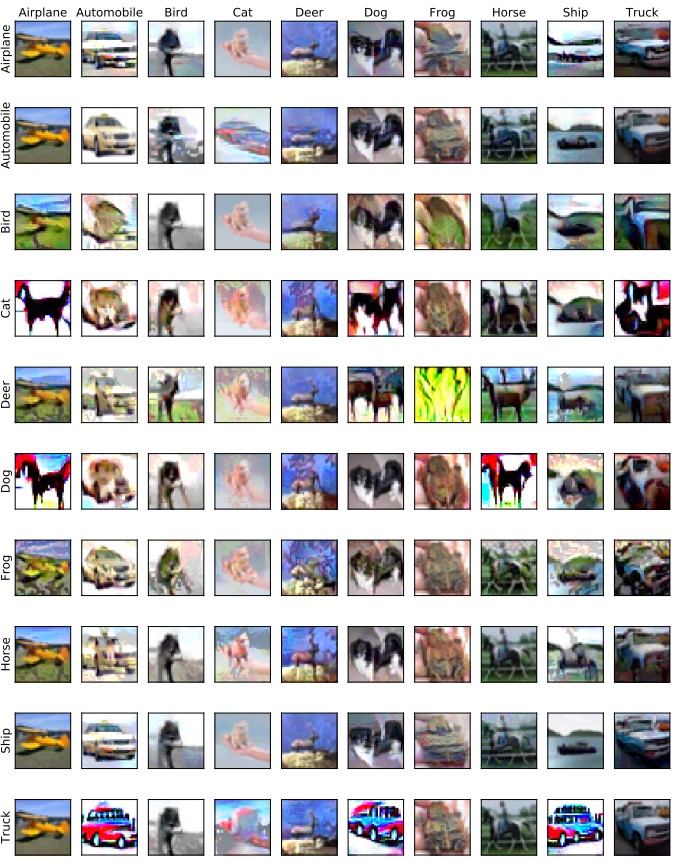

Figure 32: targeted attack visualization for **L** trained with CIFAR10. The titles of the columns describe the original class, and the titles of the rows describe the target classes.

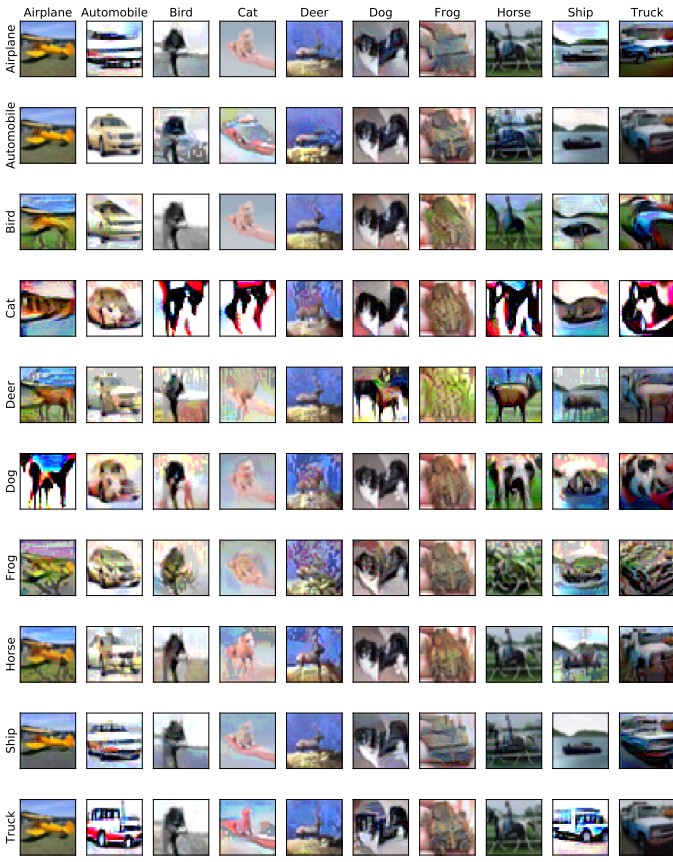

Figure 33: targeted attack visualization for **LR** trained with CIFAR10. The titles of the columns describe the original class, and the titles of the rows describe the target classes.

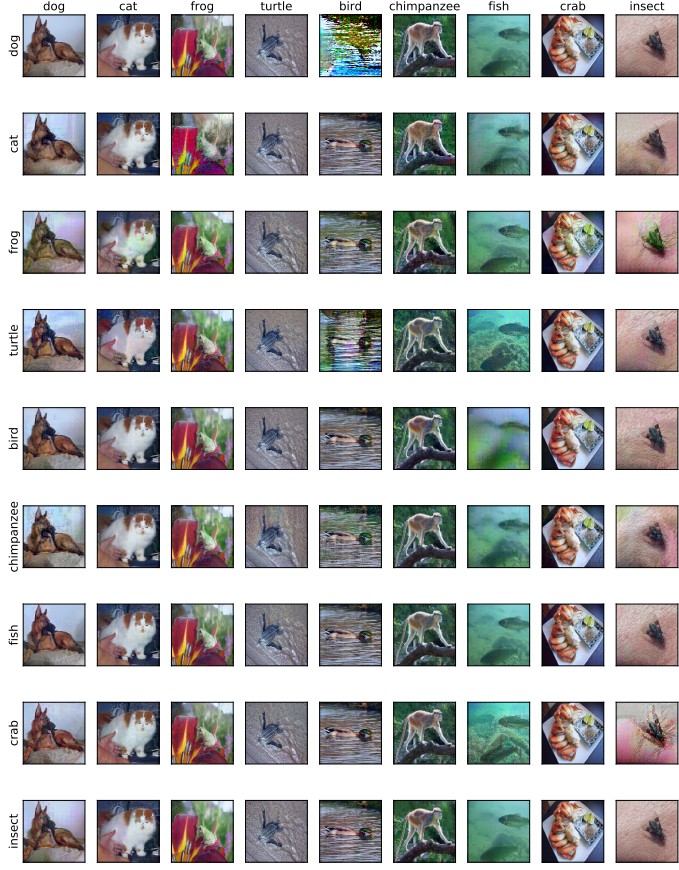

Figure 34: targeted attack visualization for **A** trained with Restricted ImageNet. The titles of the columns describe the original class, and the titles of the rows describe the target classes.

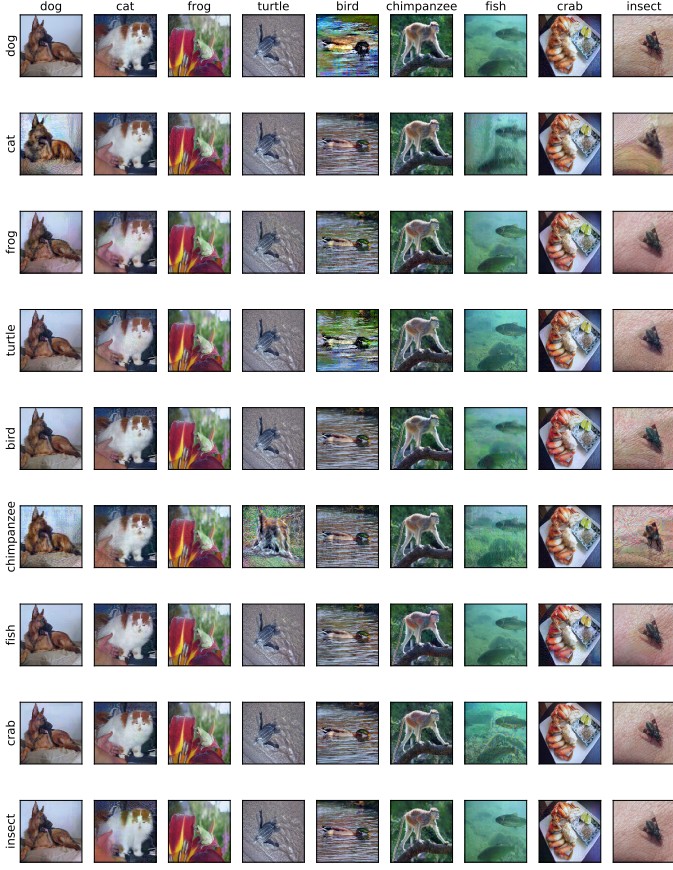

Figure 35: targeted attack visualization for **AR** trained with Restricted ImageNet. The titles of the columns describe the original class, and the titles of the rows describe the target classes.

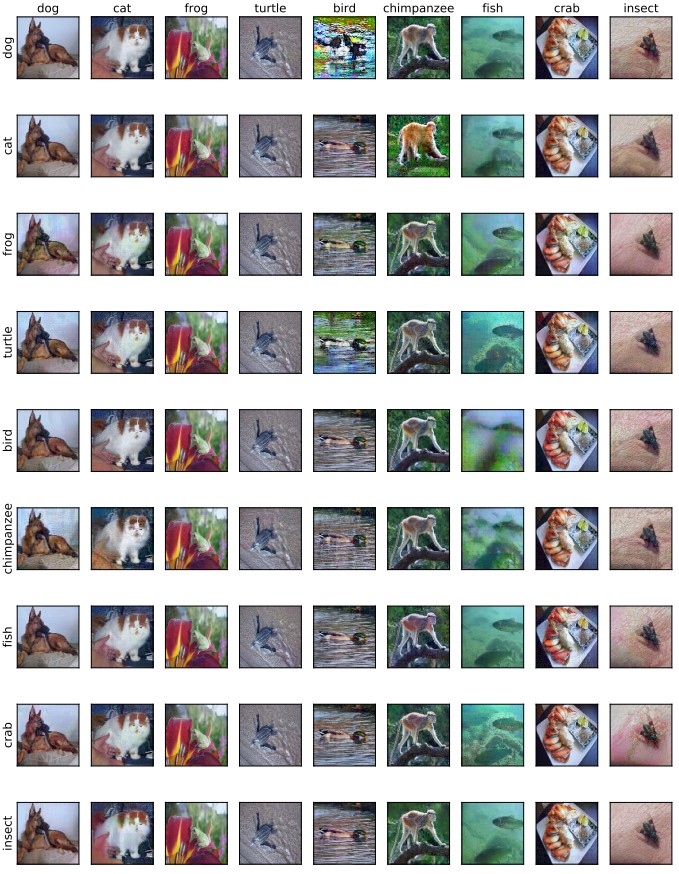

Figure 36: targeted attack visualization for **L** trained with Restricted ImageNet. The titles of the columns describe the original class, and the titles of the rows describe the target classes.

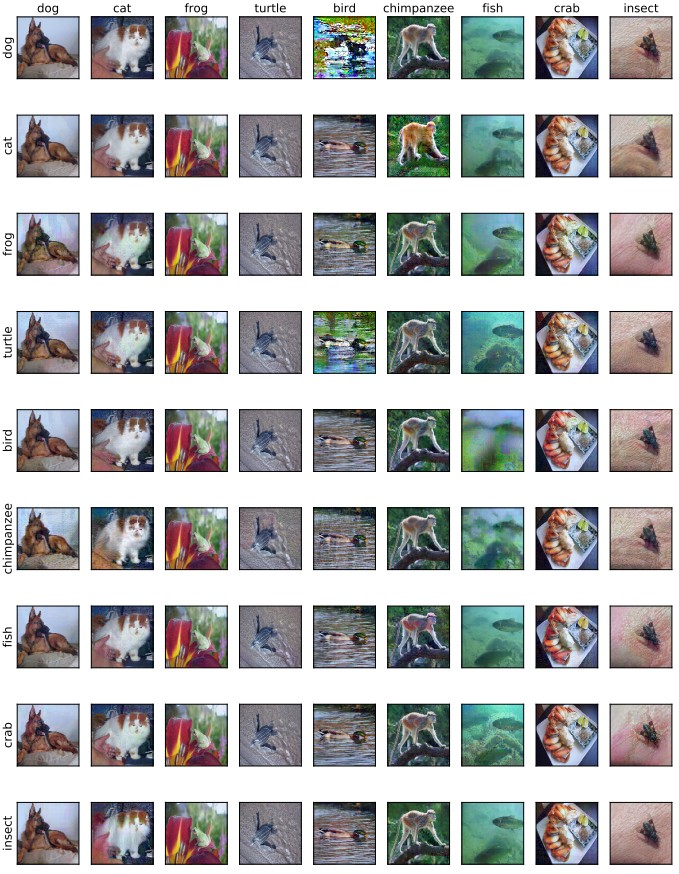

Figure 37: targeted attack visualization for **LR** trained with Restricted ImageNet. The titles of the columns describe the original class, and the titles of the rows describe the target classes.

# D  SELECTIVE ADVERSARIAL EXAMPLES

We visualize the generated adversarial examples to help us evaluate the models. We visualize the on-average most deceptive examples (the highest prediction confidence on the wrong class). We plot one example for each class of the data. For MNIST and FashionMNIST, we focus on the visualization of adversarial examples generated by Adef attack because the attack is more visually aligned with how human perceive the images.

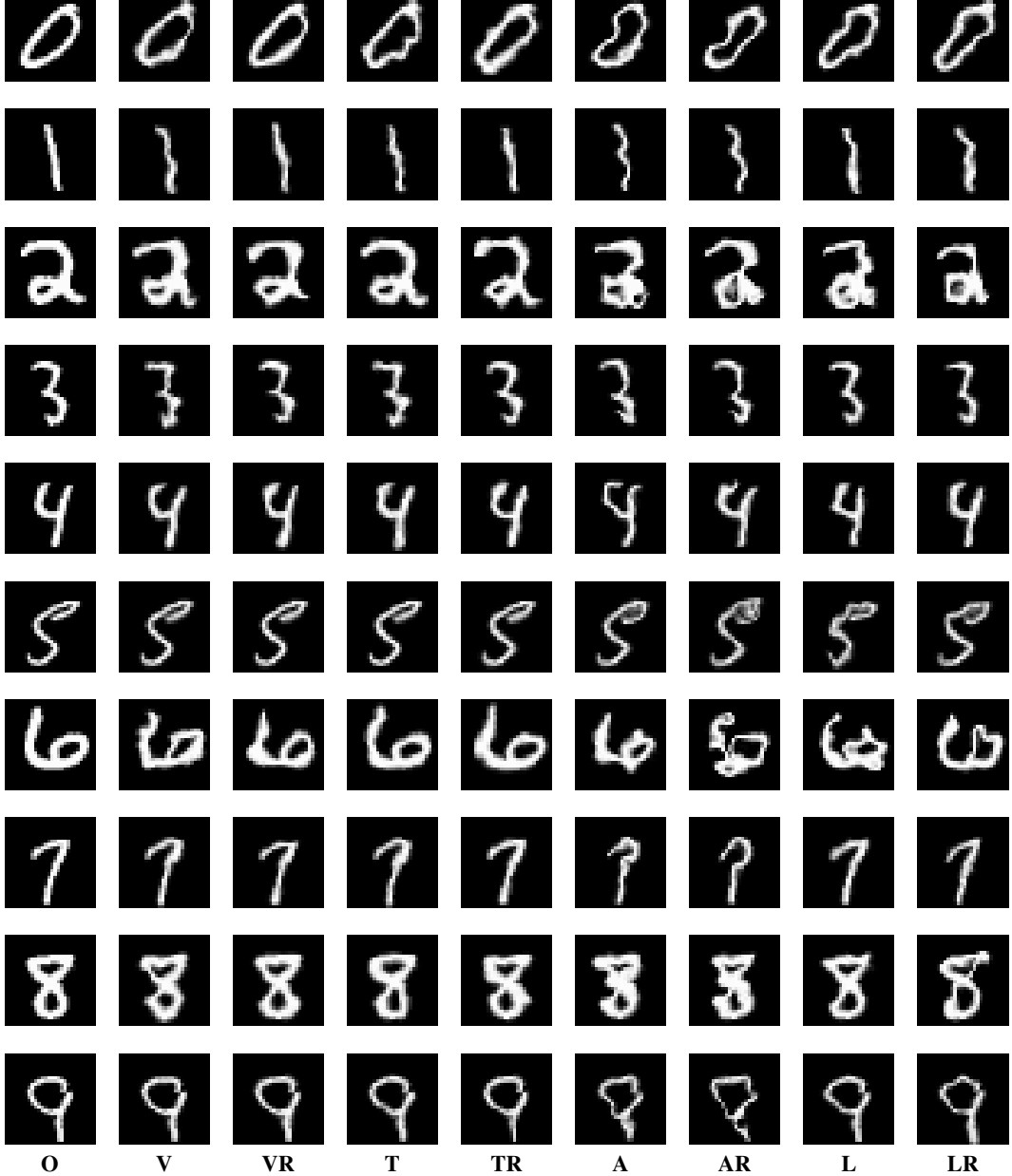

**O        V        VR        T        TR        A        AR        L        LR**

Figure 38: Adversarial Images generated by Adef attack

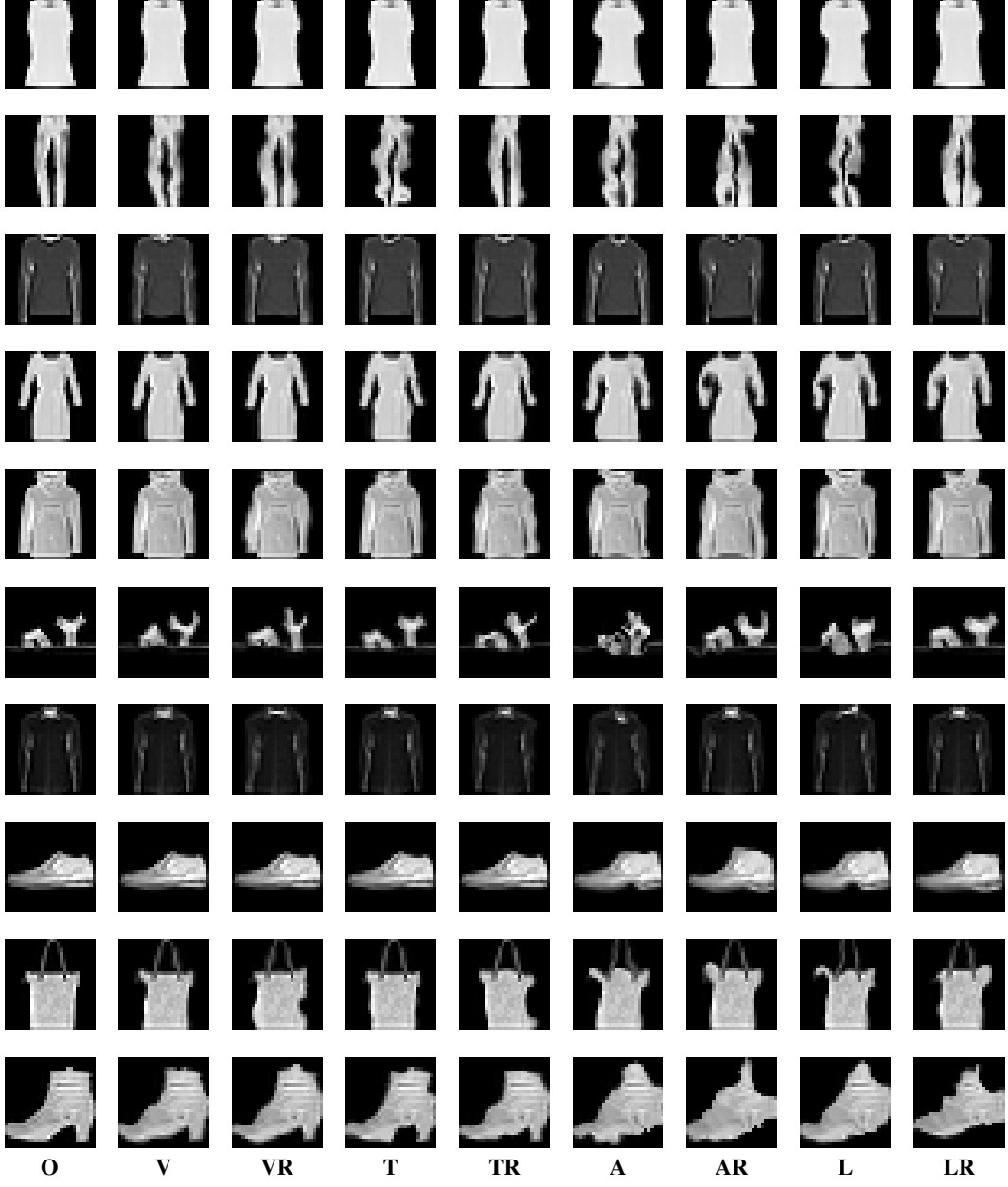

O     V     VR     T     TR     A     AR     L     LR

Figure 39: Adversarial Images generated by Adef attack

