# OpenReview forum: "Smooth Kernels Improve Adversarial Robustness and Perceptually-Aligned Gradients"
_ICLR.cc/2020/Conference — Reject_

### Official Review · AnonReviewer1 · 2019-10-21
**Official Blind Review #1**

**Rating:** 1

**Review:**

Paper summary:  This paper argues that reducing the reliance of neural networks on high-frequency components of images could help robustness against adversarial examples. To attain this goal, the authors propose a new regularization scheme that encourages convolutional kernels to be smoother. The authors augment standard loss functions with the proposed regularization scheme and study the effect on adversarial robustness, as well as perceptual-alignment of model gradients.


Comments: I will first discuss some high-level concerns I have with the paper, followed by more specific comments.

Motivation and general idea:
The idea of improving model robustness by eliminating high-frequency components in the data is not new. This has in fact been the motivation behind several (later shown to be unsuccessful) defenses such as JPEG compression. In recent work, Yin et al. explore this phenomenon in depth, and importantly discuss how adversarial examples are by no means entirely a high-frequency phenomenon. Specifically, they show that while adversarial examples for natural models do tend to be biased towards higher-frequency components, this is not true for robust models. They argue that it is in general always possible to find adversarial examples in the lower end of the frequency spectrum (for instance, transfer attacks from robust models). Thus, based on evidence from prior work, it is unlikely that a defense based entirely on removing high-frequency components from the input data can be successful.

Empirical evaluation:

There are significant issues with the empirical evaluation of the proposed defense. In particular, in Table 1 (also Figures 7-9 in the Appendix):

1. The numbers suggest that FGSM is more successful as an attack than PGD. The later is a multi-step version of the former, and hence should be strictly better (more successful in lowering model accuracy). This clearly highlights that the PGD attack is not being used correctly/not run with enough steps, and thus the numbers are not an accurate reflection of model robustness.

2. At a higher-level, the numbers that the authors highlight in the table are the best performance over attacks. This is not the correct way to evaluate robustness, which must always be reported as the *worst-case performance* of the model and hence the lowest accuracy over attacks. If one takes this into consideration, it is clear that the proposed regularization does not really improve robustness.

3. Furthermore, the authors state they use default parameters from Foolbox to evaluate their models. The issue with this is evident in the MNIST/FashionMNIST results where even with arbitrary eps, strong attacks like PGD are not able to break the model. This does not mean that the model is robust, it just means that the default hyperparameters from Foolbox cannot break the model. The authors need to re-run the attacks with different steps sizes/step counts for various attacks. The goal in evaluating model robustness should not be finding one attack (or one set of hyperparameters) which is not able to fool the model, but to show that *no attack* (for the given perturbation set) can lower model accuracy.

4. The eps that are used for the CIFAR-10 and ImageNet experiments are extremely large, and not standard in the literature. In fact, based on my experience, it is probably not possible to be (too) robust to eps as large as 0.1 on CIFAR as this is large enough to visibly change the class even for a human (cf. Tsipras et al.).

Thus, I think the robustness evaluation in Table 1 is incorrect and violates some basic sanity checks (such as PGD being stronger than FGSM), and thus does not accurately reflect the model’s true performance.


Other comments:

i. It is unclear why the sensitivity of the proposed regularization scheme to the scale of w can be fixed by subtracting the norm of w (the exp loss will just be raised to the power alpha if you scale weights by alpha). Dividing by the norm would be the more correct way to fix the scale invariance and the authors should include results with this regularization, at least in the Appendix.

ii. In Figure 1, the kernels and activations that are visualized are after the first convolutional layer. The authors should perform a similar visualization after the second layer (or later layers in general) because it is somewhat obvious that you will get these properties at the first layer with the proposed regularization scheme (if the lambda is correctly tuned). The part that is unclear is how effective the regularization is over repeated applications of convolution coupled with non-linearities such as ReLU/pooling.

iii. While the visualizations in Figures 2-6 are interesting, it seems that many of the cases in which +R looks semantically meaningful, it also looks meaningful with the base loss itself. It thus is unclear whether the perceptual alignment is coming from the base loss, or from the added regularization. Moreover, the perceptual alignment of models should improve when their dependance on “human-meaningless” features reduces. Since high-frequency components are likely “human-meaningless” features, it is plausible that the proposed regularization scheme makes activation maximization/gradients more perceptually alignment. However, I think this is somewhat orthogonal to (and is not enough to tell us anything about) the robustness of the model itself, which is affected by its reliance on many kinds of “non-robust” features (Ilyas et al., 2019) of which high-frequency components might just be one example.

iv. In recent work by Wang et al., the authors conduct an interesting experiment to see how dependent the prediction of a given classifier is to high-frequency components in the data (vs low frequency components) (cf. Figure 1 in their paper). It would be interesting to see this experiment replicated with the proposed regularization scheme.


Overall, I think there are significant issues with the paper, especially in the empirical evaluation section. The authors need to re-evaluate their models with the strongest possible form of the attack and demonstrate an improvement in worst-case performance to actually establish the merits of the proposed regularization scheme. Thus, I recommend rejection.


References:
Yin, Dong, et al. "A fourier perspective on model robustness in computer vision." arXiv preprint arXiv:1906.08988 (2019).

Tsipras, Dimitris, et al. "Robustness may be at odds with accuracy." arXiv preprint arXiv:1805.12152 (2018).

Ilyas, Andrew, et al. "Adversarial examples are not bugs, they are features." arXiv preprint arXiv:1905.02175 (2019).

Wang, Haohan, et al. "High Frequency Component Helps Explain the Generalization of Convolutional Neural Networks." arXiv preprint arXiv:1905.13545 (2019).



**Experience Assessment:**

I have published in this field for several years.

**Review Assessment: Checking Correctness Of Derivations And Theory:**

N/A

**Review Assessment: Checking Correctness Of Experiments:**

I carefully checked the experiments.

**Review Assessment: Thoroughness In Paper Reading:**

I read the paper thoroughly.

---

### Official Review · AnonReviewer3 · 2019-10-23
**Official Blind Review #3**

**Rating:** 1

**Review:**

The authors propose a method for learning smoother convolutional kernels with the goal of improving robustness and human alignment. Specifically, they propose a regularizer penalizing large changes between consecutive pixels of the kernel with the intuition of penalizing the use of high-frequency input components. They evaluate the impact of their method on the adversarial robustness of various models and class visualization methods.

At a high level, the proposed idea is interesting. Reducing the reliance of classifiers on high-frequency patterns is a plausible way of improving human alignment. However, the experimental evidence presented is either unreliable or not sufficient to demonstrate the merit of the approach.

My biggest concern is the evaluation of adversarial robustness. The authors perform a number of off-the-shelf attacks using the foolbox library without accounting for fundamental differences between these attacks or basic principles of adversarial evaluation. Specifically, when evaluating the robust accuracy of a model, one needs to specify a concrete threat model (e.g., perturbations of L2-norm at most X), perform various attacks respecting this threat model, and report the number of  examples correctly classified against _all_ attacks (see https://arxiv.org/abs/1902.06705 for additional discussion and justification). Instead, the authors consider attacks with unbounded perturbation (for which robust accuracy does not make sense) and attacks using different perturbation constraints (ADef), while reporting improvements on a per-attack base. Hence, most of the columns of Table 1 are unreliable and cannot be taken into account when evaluating the models' robustness.

Even ignoring these issues, taking into account the only column of Table 1 that is indicative, the PGD attack, we do not see sufficient evidence for the merit of the method. With the exception of a single outlier for MNIST TR (which I cannot interpret), the proposed regularizer only marginally affects the robustness of a classifier. In particular, no models become robust by adding the regularizer and the impact is limited for the case of already robust models.

Furthermore, the evidence in favor of human alignment is relatively weak. While the method does have some effect for the case of simple models (MNIST and Fashion-MNIST), for the case of complex models (for which visualization is actually a challenging problem) the improvement is virtually non-existent. For Figures 5 and 6, the columns with R have essentially the same visual quality as the original columns. The only exception are the L and A columns for CIFAR10 bird and deer for which the original visualization fails. However, I find this confusing. There exist several works by now performing visualization using adversarial models and I have never encountered such a failure before. I wonder if there is some issue with the training of these particular models.

Overall, while the high-level idea of the paper is interesting, the experimental evidence presented is either weak or unreliable. I will thus recommend rejection.

**Experience Assessment:**

I have published in this field for several years.

**Review Assessment: Checking Correctness Of Derivations And Theory:**

I carefully checked the derivations and theory.

**Review Assessment: Checking Correctness Of Experiments:**

I carefully checked the experiments.

**Review Assessment: Thoroughness In Paper Reading:**

I read the paper thoroughly.

---

### Official Review · AnonReviewer2 · 2019-10-24
**Official Blind Review #2**

**Rating:** 1

**Review:**

1.	Please number the equations for better readability.
2.	The usage of smooth kernel regularization, R(w) and logit pairing equation seems disjoint, what is the natural flow here?
3.	The notation <X, Y> is misleading, do not use inner product notation to denote tuple!
4.	I do not see any justification of using KL
5.	The contribution in terms of designing R(w) and the logit pair loss function is trivial.
6.	The argmax to get x’ how practically you ensure the solutions are inside the closed ball? The authors mentioned about robustness without any justification. Why related to TV norm? The usage of Pinsker’s inequality to get upper bound is meaningful but that doesn’t prove the robustness. Please explain, also why not state this as a theorem. I suggest prove the robustness in a concrete manner, maybe using influence functions.
7.	What is the loss function l(.,.) supposed to be?
8.	In R(w) the second part supposed to minimize the norm of convolution kernel I presume, then why there is a negative sign!
9.	The title of Section 4.1 is pretty strange, you don’t have to say “… for sanity checking”.
10.	The synthetic experiment is very immature and inconclusive. What can we get from Table 1? Also please justify the choices of hyperparameters used.
11.	Again, looking at Fig. 4, I really can’t see the usefulness.
12.	Section 4.3 is meaningless and seems redundant. Why not have a single story rather than so many branching, the experiments are not convincing at all.
13.	In Fig. 5, authors argued AR is better than A, I don’t see why, e.g., for horse A looks uch better than AR, same for insect.
14.	The regularization seems not that useful, to me this work tries to justify using a regularization which by the choice of experiments is not well grounded.


**Experience Assessment:**

I have published one or two papers in this area.

**Review Assessment: Checking Correctness Of Derivations And Theory:**

I carefully checked the derivations and theory.

**Review Assessment: Checking Correctness Of Experiments:**

I carefully checked the experiments.

**Review Assessment: Thoroughness In Paper Reading:**

I read the paper thoroughly.

---

### Decision · Program_Chairs · 2019-12-19

**Decision:**

Reject

**Comment:**

The authors propose a regularized for convolutional kernels that seeks to improve adversarial robustness of CNNs and produce more perceptually aligned gradients. While the topic studied by the paper is interesting, reviewers pointed out several deficiencies with the empirical evaluation that call into question the validity of the claims made by the authors. In particular:

1) Adversarial evaluation protocol: There are several red flags in the way the authors perform adversarial evaluation. The authors use a pre-defined adversarial attack toolbox (Foolbox) but are unable to produce successful attacks even for large perturbation radii - this suggests that the attack is not tuned properly. Further, the authors present results over the best case performance over several attacks, which is dubious since the goal of adversarial evaluation is to reveal the worst case performance of the model.

2) Perceptual alignment: The claim of perceptually aligned gradients also does not seem sufficiently justified given the experimental results, since the improvement over the baseline is quite marginal. Here too, the authors report failure of a standard visualization technique that has been successfully used in prior work, calling into question the validity of these results.

The authors did not participate in the rebuttal phase and the reviewers maintained their scores after the initial reviews.

Overall, given the significant flaws in the empirical evaluation, I recommend that the paper be rejected. I encourage the authors to rerun their experiments following the feedback from reviewers 1 and 3 and resubmit the paper with a more careful empirical evaluation.